# CN_Wheat10: A 10 m resolution dataset of spring and winter wheat distribution in China (2018–2024) derived from time-series remote sensing

Man Liu[1], Wei He[1], Hongyan Zhang[1, 2]

[1]State Key Laboratory of Information Engineering in Surveying, Mapping and Remote Sensing, Wuhan University, Wuhan, 430079, PR China.
[2]School of Computer Science, China University of Geosciences, Wuhan, 430074, PR China

*Correspondence to*: Hongyan Zhang (zhanghongyan@cug.edu.cn)

**Abstract.** Wheat, as one of the main food crops in the world, plays a vital role in shaping agricultural trade patterns. China is the largest producer and consumer of wheat globally, characterized by extensive cultivation areas and diverse planting systems. However, current remote sensing-based wheat mapping studies often rely on uniform phenological feature variables, without adequately accounting for the significant differences in wheat growth cycles across China's diverse agro-ecological zones. In addition, the lack of large-scale training samples severely limits both the accuracy and the spatial-temporal generalization capacity. Furthermore, existing research in China focuses mainly on winter wheat monitoring and mapping, while spring wheat research remains largely inadequate, especially in the major spring wheat-producing areas of northern China, resulting in limited availability of targeted remote sensing products. These limitations hinder the development of high-accuracy, spatially comprehensive wheat mapping datasets and reduce the completeness of agricultural monitoring and food security assessments. To address these issues, this study proposes a cross-regional training sample generation method that integrates time-series remote sensing data with crop distribution products. Furthermore, a province-level, differentiated feature selection strategy is introduced to enhance the regional adaptability and classification performance of the model. Based on these methods, we developed 10 m resolution wheat distribution dataset (CN_Wheat10) covering the years 2018–2024. The dataset includes spring and winter wheat harvested area maps for 15 provinces and detailed winter wheat planted area maps for 10 provinces across China. Validation using a large-scale reference dataset built from field surveys and high-resolution imagery visual interpretation indicates that CN_Wheat10 achieves mapping accuracies above 0.93 for winter wheat and above 0.91 for spring wheat. When compared with wheat area statistics from the China Statistical Yearbook, the coefficient of determination ($R^2$) exceeds 0.94 at the provincial level and remains above 0.88 at the municipal level. Spatially, wheat cultivation in China is characterized by a pattern of concentration in the east, dispersion in the west, a dominance of winter wheat, and a supplementary role of spring wheat. CN_Wheat10 provides spatial distribution information on both spring and winter wheat harvested areas and winter wheat planted regions in key production areas. Compared with existing products that mainly focus on winter wheat, this dataset expands both the spatial coverage and the crop types, offering more comprehensive data support for agricultural monitoring and management in China. The CN_Wheat10 product is freely accessible at https://doi.org/10.6084/m9.figshare.28852220.v2 (Liu et al., 2025a).

# 1 Introduction

Wheat, as one of the world's three major staple crops, holds an irreplaceable strategic role in maintaining social stability (Singh et al., 2023). As the largest wheat producer and consumer globally, China has consistently ranked among the top in annual wheat output, serving both as a cornerstone of national food security and an important player in global grain trade regulation (Dong et al., 2024). In recent years, driven by population growth, dietary transitions, and increasing demand from the livestock sector, domestic wheat consumption in China has continued to rise. Despite a relatively high self-sufficiency rate, China still engages in wheat import and export to optimize variety structure and supplement high-quality grain supply. Currently, global climate change has caused a rise in the occurrence of extreme weather events, while geopolitical conflicts have triggered fluctuations in international food markets, posing dual threats to the stability of wheat production and the security of trade chains (Li and Song, 2022; Tilman et al., 2011). Against this backdrop, developing a high-accuracy, wide-coverage remote sensing monitoring system for wheat, and achieving nationwide, high spatiotemporal resolution mapping, is not only a technical foundation for advancing precision agriculture, but also a critical component for strengthening early warning and emergency response capabilities in national food security.

The continuous evolution of remote sensing technology has made satellite imagery indispensable for agricultural monitoring (Dong et al., 2024; Blickensdörfer et al., 2022). In particular, for large-scale and cross-regional crop mapping tasks, the implementation of automated and standardized workflows based on satellite imagery has proven critical for the timely acquisition and dynamic updating of agricultural datasets (Lin et al., 2022; Ghassemi et al., 2024). Currently, several international organizations and governmental agencies have developed publicly accessible crop mapping products, some of which incorporate dedicated layers for wheat. For instance, the European Crop Type Map at 10 m resolution leverages Sentinel imagery to enable fine-scale mapping of major crop types across Europe, including key staples such as wheat (D'andrimont et al., 2021). In the United States, the Cropland Data Layer (CDL) has become the most authoritative and widely used crop mapping product, with consistently high accuracy in wheat mapping (Boryan et al., 2011). Similarly, Statistics Canada provides 30 m Annual Crop Inventory product, which covers the entire agricultural zone of the country and includes multiple crop types (Amani et al., 2020). These crop products not only support domestic agricultural policy formulation and scientific research, but also serve as valuable benchmarks for the development and validation of crop mapping methodologies at the global scale.

China is among the world's top wheat producers, boasting extensive cultivation areas and diverse cropping systems nationwide (Mottaleb et al., 2023; Dong et al., 2024; Tao et al., 2012). Due to variations in climatic and geographical conditions, winter wheat and spring wheat exhibit significant differences in phenology, climatic adaptability, and spatial pattern. Winter wheat is predominantly cultivated in the eastern plains, while spring wheat is mainly grown in the northwest and northeast regions (Liu et al., 2018; Zhang et al., 2022b). Several researches have conducted thematic mapping of wheat distribution in China, resulting in remote sensing-based wheat products with relatively high spatial resolution. For instance, some studies have employed the Time-Weighted Dynamic Time Warping method combined with time-series imagery to produce 30 m

winter wheat product in China from 2001 to 2023 (Dong et al., 2020). Other studies have used phenology-based algorithms to generate 30 m winter wheat product across 11 provinces from 2001 to 2020 (Dong et al., 2024). Additionally, researchers have utilized spectral phenological features and elevation data to map winter wheat planted and harvested areas from 2018 to 2022 (Hu et al., 2024). Another approach integrated winter wheat phenology, spectral, and polarization characteristics into sample generation methods, combined with Random Forest (RF) algorithm, to produce 10 m winter wheat product between 2018 and 2024 (Yang et al., 2023). Other studies combined Sentinel-1/2 data to map wheat planting patterns in China in 2020, including the distribution of spring and winter wheat and rotation patterns (Qiu et al., 2025). However, these existing studies and publicly available products have primarily focused on the mapping of winter wheat, with limited attention to the systematic characterization of spring wheat distribution. As a key staple crop in northwest and northeast China, spring wheat accounts for a certain portion of the national wheat production system. Neglecting spring wheat leads to incomplete representation in remote sensing-based wheat mapping. Moreover, most current mapping approaches adopt uniform spectral features across the entire country, without fully accounting for regional differences in phenological patterns, climatic conditions, and agricultural practices. This lack of regional adaptability limits the accuracy of wheat products.

Throughout the crop growth cycle, a range of environmental and human factors can affect development from planting to harvest, often causing noticeable differences in both time and space between the sown area and the area actually harvested (Wei et al., 2023; Baker et al., 2019). Wheat is typically sown during periods with favorable climatic conditions to ensure successful germination and early growth. However, during subsequent growth stages, certain regions may be subject to environmental challenges such as drought, prolonged heat, or pest and disease outbreaks, potentially leading to yield reduction, premature senescence, or even total crop failure (Wu et al., 2021; Tao et al., 2022). While remote sensing can effectively identify wheat planting areas at large scales, some fields may ultimately fail to be harvested due to poor yield performance or complete crop loss. Consequently, the final harvested area often falls short of the area originally planted. According to agricultural statistics from the United States, crop harvest rates were generally below 85% between 1970 and 2017 (Zhu and Burney, 2021). Similarly, in China, the harvested area of winter wheat between 2018 and 2022 was approximately 12.88% lower than the planted area (Hu et al., 2024). Therefore, remote sensing-based mapping that encompasses both the planted and harvested area of wheat is essential not only for improving the timeliness and accuracy of crop distribution identification, but also for providing early warning information to agricultural management authorities. Such approaches enhance the capacity to detect potential yield losses and contribute to the advancement of food security management toward more refined and intelligent decision-making frameworks.

Mainstream methods for wheat mapping using remote sensing largely rely on spectral phenology, often supported by machine learning algorithms to boost precision and adaptability (Ashourloo et al., 2022; Xie and Niculescu, 2022; Hu et al., 2019). Spectral phenology-based methods exploit the distinct multispectral reflectance characteristics of different types and utilize phenological curves over the crop growth cycle to enable dynamic crop type identification. These methods are particularly effective for crops such as winter wheat, which exhibit relatively stable and predictable phenological patterns.

Several studies have extracted key phenological characteristics from winter wheat growth curves to identify spatial distribution (Qu et al., 2021; Tao et al., 2017; Fu et al., 2025), while others have designed mapping indices based on the temporal variation between stages (Qiu et al., 2017; Yang et al., 2023). However, the effectiveness of these approaches is contingent upon the temporal consistency of remote sensing imagery, which can be significantly compromised by cloud cover and discontinuities in data acquisition. The integration of spectral phenological features with machine learning methods allows for the fusion of multi-source feature information and supports automated learning of the spatiotemporal distribution patterns of wheat, significantly improving model generalization and robustness. For instance, some studies have successfully applied time-series Sentinel-1/2 imagery in combination with the RF algorithm to map winter wheat across multiple countries (Yang et al., 2024). Others have employed deep learning models and time-series imagery to accurately delineate wheat production systems in eight countries worldwide (Luo et al., 2022). While spectral phenology provides a solid data foundation for wheat identification, and machine learning offers strong adaptability in large-scale and topographically complex regions, these strategies are highly dependent on the presence of accurate field-validated samples. Acquiring such samples typically requires time-consuming and labor-intensive field surveys. Therefore, in the development of national-scale wheat remote sensing products, the construction of reliable sample datasets and the integration of multi-feature information that accounts for regional variability are critical to achieving high-accuracy wheat mapping.

To address the aforementioned challenges, this study developed a systematic sample generation strategy and a province-level feature selection approach for wheat mapping, and subsequently produced a remote sensing monitoring dataset of wheat in China, named CN_Wheat10. This dataset covers 15 provinces from 2018 to 2024 and was generated from time series Sentinel images. By integrating multiple spectral and phenological features, CN_Wheat10 accounts for the region-specific spatial layouts of both spring and winter wheat, and includes information on both planted and harvested areas. First, spring and winter wheat training samples applicable to China were constructed using U.S. remote sensing imagery and the CDL product. Second, a region-specific feature selection strategy was implemented to accommodate the phenological differences of wheat across provinces, thereby improving mapping accuracy. Third, relying on the Google Earth Engine platform, annual large-scale wheat distribution maps were generated with high timeliness and spatial resolution. Finally, the resulting dataset was systematically evaluated using extensive manually validated samples, existing public products, and agricultural statistical data. Compared to existing wheat remote sensing products, CN_Wheat10 expands the spatial coverage and provides a more detailed understanding of wheat's spatial distribution across China.

## 2 Study area and data

### 2.1 Study area

The study area (Fig. 1) encompasses 15 provinces and 3 municipalities in China, including Anhui (AH), Gansu (GS), Hebei (HB), Henan (HN), Hubei (HuB), Jiangsu (JS), Inner Mongolia (NM), Ningxia (NX), Qinghai (QH), Shandong (SD), Shanxi

(SX), Shaanxi (SAX), Sichuan (SC), Xinjiang (XJ), Zhejiang (ZJ), Beijing (BJ), Tianjin (TJ), and Shanghai (SH). In 2022, these provinces and municipalities accounted for approximately 97.8% of China's total wheat area and 99% of wheat production (https://www.stats.gov.cn/sj/ndsj/). Given the relatively small administrative areas of the municipalities and the strong spatial continuity of their agricultural zones with adjacent provinces, appropriate regional adjustments were made during the mapping process. Specifically, BJ and TJ were integrated into the HB province mapping zone, while SH was merged with JS province. Among the 15 provinces included in this study, the ten provinces located in eastern and southern China, which encompass the Huang–Huai–Hai Plain and the middle and lower reaches of the Yangtze River Plain, constitute the country's core winter wheat production regions. These areas are characterized by large cultivation scales and highly contiguous fields, which enables the extraction of both planted area maps and harvested area maps. In contrast, in the five provinces located in northern and northwestern China, wheat cultivation is relatively limited and fragmented, and some regions involve mixed planting of spring wheat and winter wheat with harvest periods that occur close to each other. As a result, only harvested area maps were generated for these provinces in this study.

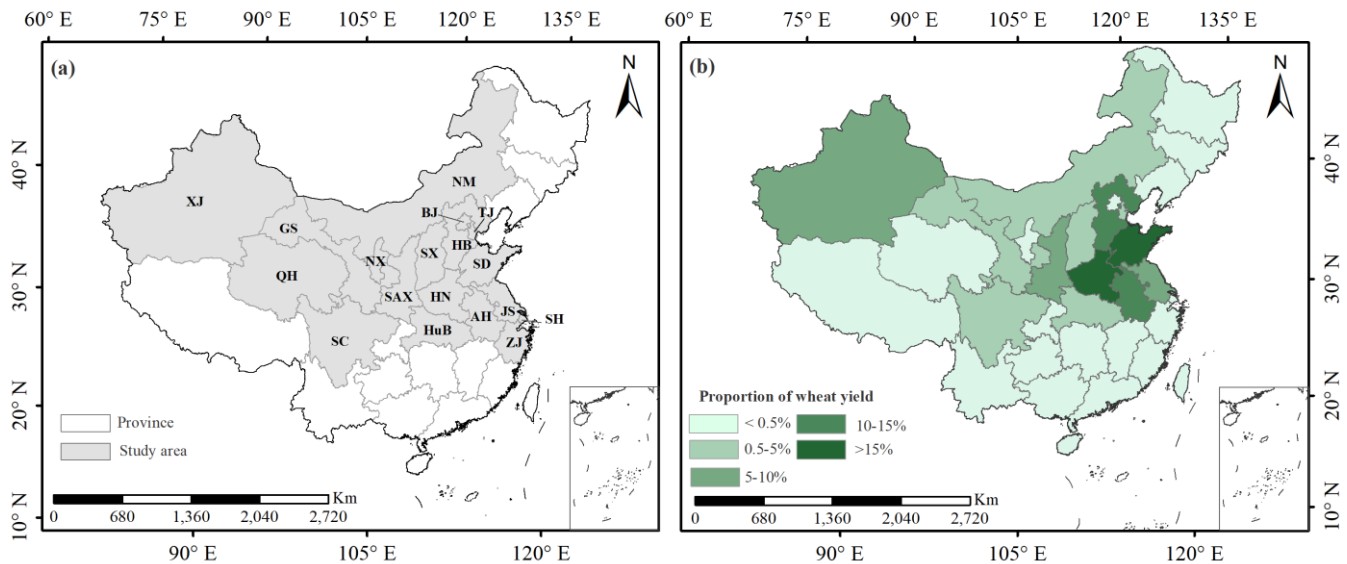

**Figure 1: Location of study area in China. (a) Location of 15 provinces and 3 municipalities included in the study area. (b) Proportion of wheat production in 2022.**

## 2.2 Study data

### 2.2.1 Remote sensing data

Sentinel-2 imagery, with rich spectral information, is particularly well-suited for large-scale crop monitoring (Xu et al., 2024a; Fan et al., 2024). In this study, 10 spectral bands with spatial resolutions of 10 m and 20 m were selected to balance spectral completeness with data processing efficiency (Xu et al., 2024b). Furthermore, 15 typical spectral indices were extracted based

on the original bands, with detailed information provided in Table S1. To complement the limitations of optical remote sensing, Sentinel-1 data were also incorporated, leveraging its capability to penetrate cloud cover and complex surface conditions to support the extraction of spatiotemporal dynamics of wheat growth (Qiu et al., 2025). For winter wheat mapping, images acquired from October in the current calendar year to June of the subsequent year were used, while spring wheat mapping

utilized imagery from April to August each year. To enhance the quality and stability of the time-series data, the Google Earth Engine (GEE) was employed. First, Sentinel-2 data with cloud cover exceeding 60% were excluded to improve overall data quality. Subsequently, cloud masking was performed on the remaining imagery using the QA60 band and the MSK_CLDPRB cloud probability band to effectively remove residual cloud contamination. A stable and continuous time series was generated from the cloud-filtered data through linear interpolation (Qiu et al., 2025). Utilizing the above-stated remote sensing imagery,

spatial distribution dataset of spring and winter wheat was generated at 10 m resolution for the years 2018 to 2024. This dataset is called CN_Wheat10 for short.

### 2.2.2 Cropland Data Layer

The Cropland Data Layer (CDL) is a high-resolution crop mapping product and covers the primary agricultural regions of the United States (Boryan et al., 2011; Hao et al., 2020). In addition to providing pixel-level mapping of major crop types, the

CDL also includes a confidence layer, which represents the classification confidence score for each pixel and indicates the reliability of the assigned label (Liu et al., 2004). In this study, the CDL products from 2018-2024 were used to generate training samples for China wheat mapping. Given the similarities in climate and cropping systems, Kansas and North Dakota were selected as representative regions for winter wheat and spring wheat, respectively.

### 2.2.3 Validation sample set

The wheat validation dataset was constructed by integrating field survey data with visually interpreted results from high-resolution remote sensing imagery. Extensive field surveys were conducted from 2020 to 2024. During these processes, the GPS-Video-GIS (GVG) mobile application was used to collect georeferenced validation samples (Wu and Li, 2012; Yang et al., 2025), including land cover types and coordinates, with approximately 2000-3000 field survey sample points per year. In addition to field data, visual interpretation was employed to supplement and enhance the validation dataset (Zheng et al., 2021).

The sampling design of validation samples refers to the methods in previous studies (Liu et al., 2024b; Liu and Zhang, 2023), and has been adjusted according to the specific conditions of this study to ensure its scientific and rationality. Multi-temporal Sentinel-2 imagery from 2017 to 2024 was dynamically explored through the Google Earth Engine (GEE) visualization platform. Manual interpretation was conducted by combining spectral, textural, and temporal variation characteristics. A spatially stratified sampling strategy based on quadrilateral grids was adopted to mitigate the effects of spatial autocorrelation.

To further improve interpretation accuracy and boundary delineation, historical very high-resolution imagery (GE-VHR) from Google Earth was used for auxiliary verification. Based on the above approach, more than 50,000 valid sample points were collected annually within the study area, covering diverse ecological zones and cropping systems. These samples included

spring wheat, winter wheat, and non-wheat land cover types, ensuring comprehensive representation across different growing conditions and regional planting patterns. The details of the validation point data are introduced in Text S1, the spatial distribution of the field survey samples is illustrated in Fig.S1, and the process of visual interpretation of the validation points is shown in Fig.S2 and Fig.S3. The provincial distribution of wheat validation points is detailed in Table S2.

### 2.2.4 Other datasets

We used provincial- and municipal-level wheat area statistics from the China Statistical Yearbook as reference data. The CN_Wheat10 product was compared with the corresponding statistical records on a year-by-year basis. Specifically, complete provincial-level data were available for the period 2018–2023, while complete municipal-level data were available for 12 provinces from 2018 to 2022. To quantify the agreement between the estimates and the statistical data, the coefficient of determination ($R^2$) was employed as the accuracy assessment metric (Liu et al., 2024b). The accuracy and temporal consistency of the CN_Wheat10 dataset were evaluated by comparing it against publicly available, high-resolution wheat mapping products for China. Details of the five wheat product maps are presented in Table 1.

**Table 1: Information on the reference wheat mapping products used for comparison.**

| Wheat maps | Wheat types | Study area | Resolution | Time range | Reference |
|---|---|---|---|---|---|
| ChinaWheat10 | winter wheat | 11 provinces | 10 m | 2018-2024 | (Yang et al., 2023) |
| ChinaWheatMap10 | winter wheat | 8 provinces | 10 m | 2018-2022 | (Hu et al., 2024) |
| ChinaCP-Wheat10m | spring and winter wheat | China | 10 m | 2020 | (Qiu et al., 2025) |
| WorldCereal | spring and winter cereals | Global | 10 m | 2021 | (Van Tricht et al., 2023) |
| TWDTW_Map | winter wheat | 11 provinces | 30 m | 2001-2023 | (Dong et al., 2020) |

Note: ChinaWheatMap10 includes planted area maps (ChinaWheatMap10_P) and harvested area maps (Chinawheatmap10_H). The last product was generated by TWDTW algorithm, we call this product TWDTW_Map for short.

### 3 Methods

The process of generating the annual distribution map of wheat is shown in Fig. 2: (1) Generation of wheat samples: High-quality spring and winter wheat samples for China were generated using the CDL data and the RF algorithm. (2) Selection of provincial feature sets: Based on the separability between wheat and non-wheat types, feature separability evaluations and feature set selection were conducted for each province. (3) Generation of annual distribution map: Using the wheat samples and provincial feature sets, RF algorithms were applied on the GEE platform to generate annual wheat distribution maps for China from 2018 to 2024. (4) Accuracy assessment of wheat distribution maps: The accuracy of the generated dataset was

systematically evaluated based on large-scale manually validated samples, existing public product layers, and data from the China Statistical Yearbook.

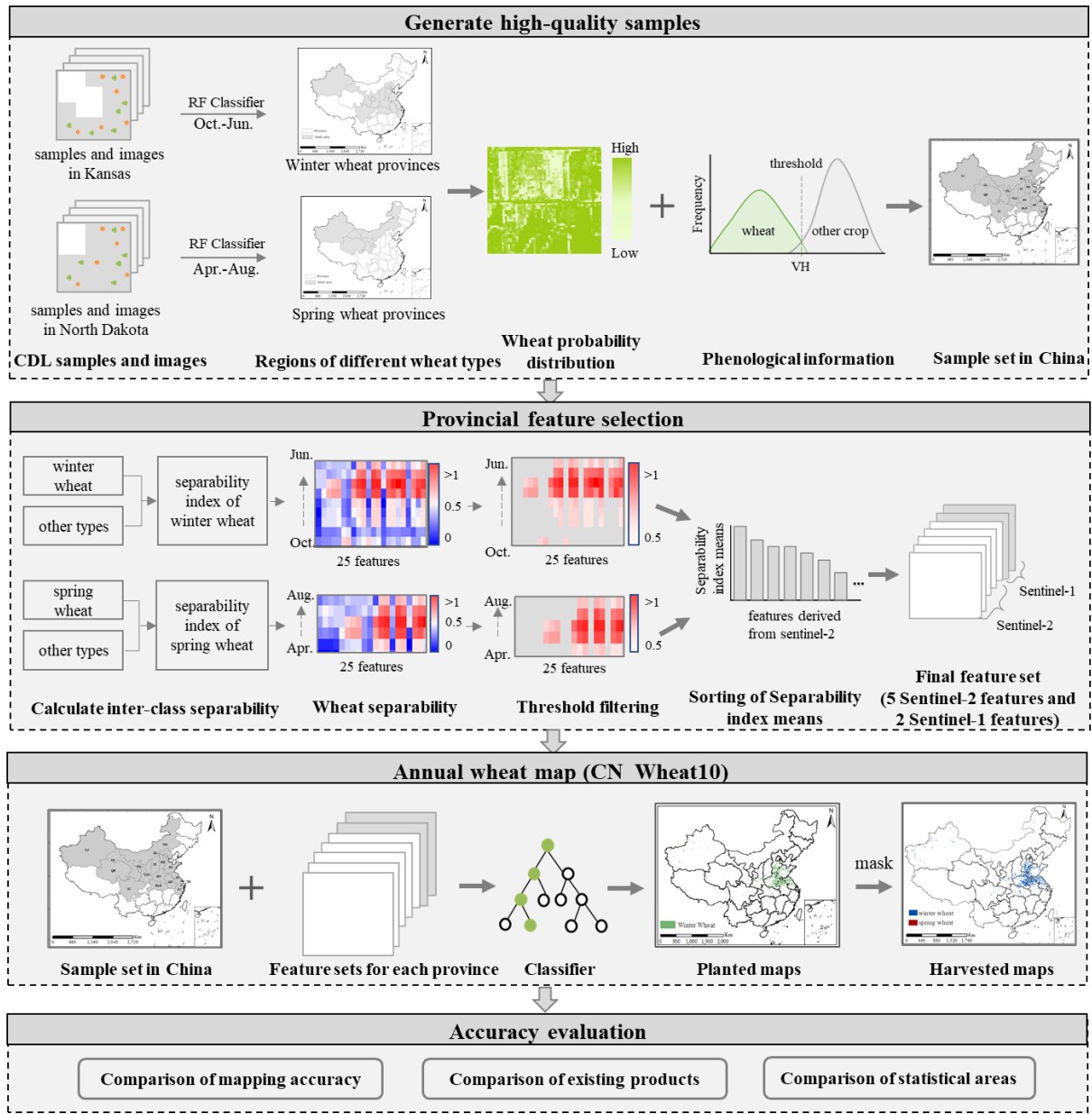

**Figure 2: Flowchart for mapping annual wheat distribution.**

To distinguish spring and winter wheat, we first predefined provinces as winter-dominant, spring-dominant, or mixed based on agronomic expertise and provincial cropping statistics. Classification workflows were then tailored accordingly. In

provinces dominated by a single crop season, only the corresponding seasonal time series was used: October–June of the following year for winter wheat and April–August for spring wheat. The resulting maps in these regions therefore represent only that season's wheat distribution, without overlap between spring and winter wheat. In mixed provinces, two independent classification chains were applied: one using winter-season imagery to detect winter wheat, and the other using spring-season imagery to detect spring wheat. Pixel-level outputs were merged based on classification probabilities, when one seasonal probability was substantially higher, the pixel was assigned to that season. This "province-level predefinition plus season-specific classification" strategy ensures consistency with dominant cropping systems while adequately capturing the complexity of mixed spring–winter wheat regions.

## 3.1 Generation of wheat samples

In this study, sample datasets suitable for spring-winter wheat regions in China were constructed using CDL data from Kansas and North Dakota, respectively, along with corresponding Sentinel-2 imagery from 2017 to 2024. Kansas and North Dakota are representative of winter and spring wheat systems, respectively, and their mid-latitude conditions result in strong phenological alignment with China's major wheat zones. As illustrated in Fig. 3, the phenological profiles from the United States closely match those of the corresponding wheat types in China, confirming the representativeness of the constructed samples.

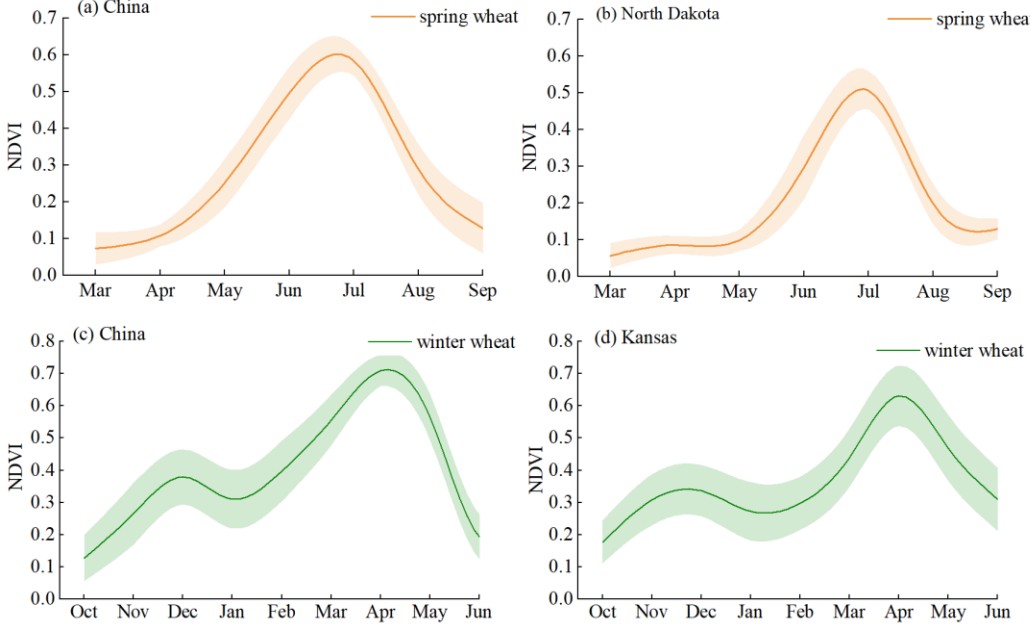

Figure 3: Comparison of NDVI time series curves between spring and winter wheat in China and the United States. (a) Spring wheat NDVI profile from field survey data in Northwest China. (b) Spring wheat NDVI profile from CDL data in North Dakota, USA. (c) Winter wheat NDVI profile from field survey data in the eastern plains of China. (d) Winter wheat NDVI profile from CDL data in Kansas, USA.

Previous studies have demonstrated that the CDL-based cross-regional approach for generating large-scale winter wheat training samples is effective and reliable in the main production areas of the Huang–Huai–Hai Plain(Liu et al., 2025b). Building upon this foundation, the present study further extends the approach to develop a comprehensive sample set that encompasses both spring and winter wheat. First, pixels with classification confidence scores greater than 95% in the CDL product were selected. A 20 km × 20 km grid-based sampling strategy was applied to extract representative wheat and non-wheat samples. These samples were then matched with multi-temporal Sentinel-2 imagery, and pixels with abnormal spectral characteristics or incomplete temporal information were removed, resulting in a high-quality source-domain sample dataset. After applying confidence filtering, grid-based sampling, and temporal matching of imagery, 5,000 samples each in Kansas and North Dakota were generated, including 2,500 for wheat and 2,500 for non-wheat. The non-wheat category includes buildings, water, fallow land, tree, grassland, and other crops. These source-domain samples were then transferred to China region using Random Forest classifier in combination with Sentinel-2 time-series imagery, thereby generating wheat probability maps for the target region.

To analyze the distribution of different land-cover types within the derived wheat probability map, we randomly selected 500 sample points for each land-cover category from the field survey data and plotted their frequency distribution against the corresponding wheat probability values. As shown in Fig. 4, confusion often occurs between wheat, rapeseed, and garlic due to similar cropping patterns, especially within the 40%–70% probability range. To improve mapping accuracy, VH-polarized backscatter coefficient from Sentinel-1 were incorporated. In calculating the VH backscatter threshold, some of the 2020 field survey data were utilized for both threshold determination and validation. It should be noted that the sample set used for this VH analysis constitutes a randomly selected subset of the overall validation samples. This approach ensures the representativeness of the analysis while using the existing high-quality ground-truth data. Figure 5 demonstrates that April is optimal for distinguishing winter wheat, while July is best for differentiating spring wheat from other spring crops. A uniform VH backscatter threshold of −17.5 dB was applied to exclude non-wheat crops within the ambiguous probability range. To evaluate the robustness of the threshold, independent samples from different years and agro-ecological zones were further tested, including: (i) Hebei Province in 2021, representing a typical winter wheat-garlic intercropping area in China; (ii) Jiangsu Province in 2022, representing a region where winter wheat coexists with winter rapeseed in China; and (iii) Qinghai Province in 2019, representing a spring wheat-spring rapeseed coexistence area in northwestern China. The results demonstrate that the threshold of –17.5 dB consistently distinguished wheat from other crops across various years and regions, confirming its robustness and transferability (Fig. S4). Finally, by integrating spatial filtering techniques with a stratified sampling strategy, a comprehensive training sample set was systematically constructed across 15 provinces in China. To ensure both regional representativeness and class balance, the number of samples in each province was determined based on a standardized grid approach, whereby each 0.5° × 0.5° grid cell was required to contain 500 sample points for wheat and 500 for non-wheat. This design effectively supports the robustness and generalizability of the classification model across heterogeneous agro-ecological zones. The sample size selection process is shown in Fig. S5.

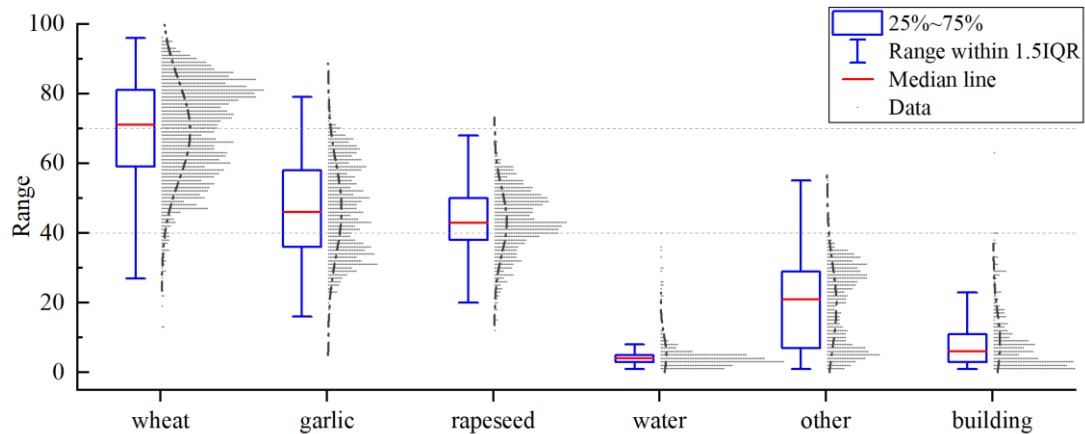

**Figure 4: Probability distribution range for different land cover types.**

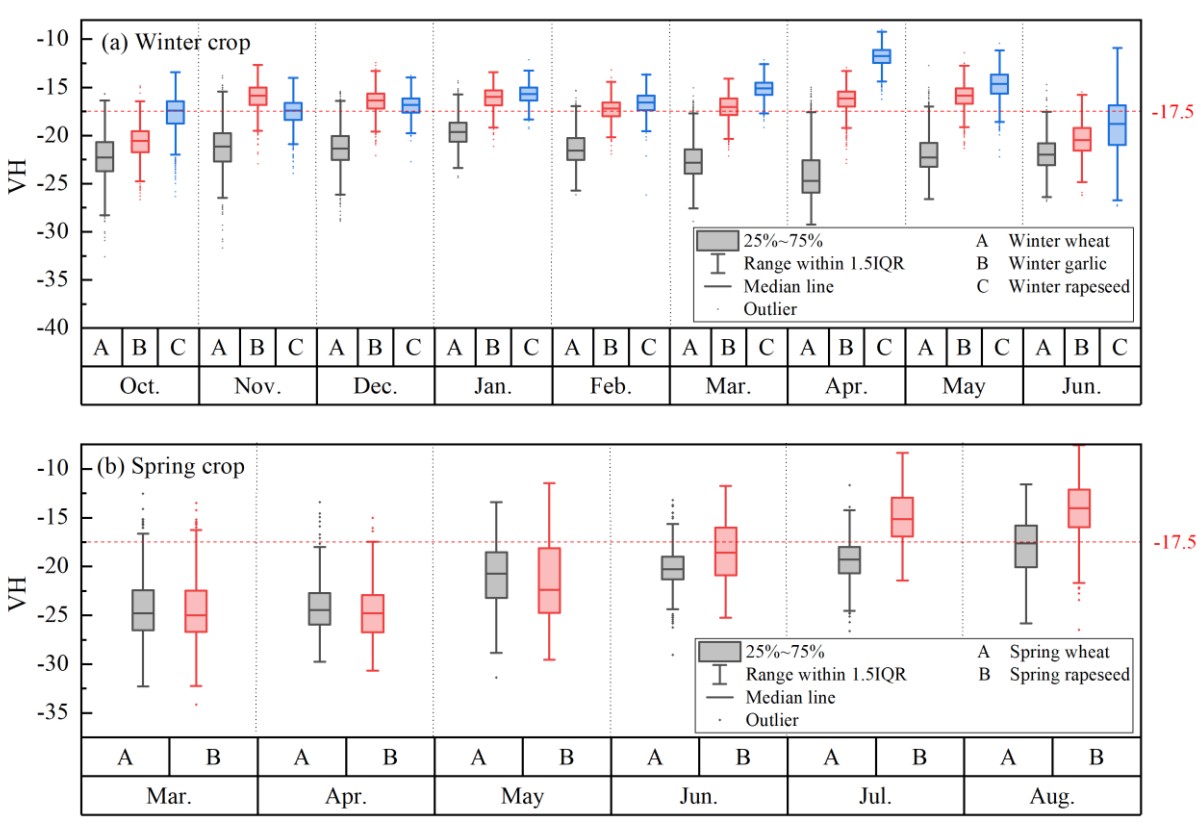

**Figure 5: VH values for different winter and spring crops.**

## 3.2 Selection of provincial feature set

To effectively reduce remote sensing mapping errors caused by phenological differences across regions, this study adopted a province-level differentiated feature selection strategy. Based on field survey samples, we examined the Normalized Difference Vegetation Index (NDVI) profiles of dominant land cover types across four provinces (Fig. 6). The results indicated that spring and winter crops exhibit distinct temporal patterns compared to other land cover types throughout their growth cycles. Winter crops mainly grow from October to June of the following year, while spring crops are mainly grown from April to August. Based on the clear differences in crop growth cycles, we designed two separate processes to distinguish between winter and spring crops. In Section 3.1, we have distinguished between spring wheat and winter wheat pixels, and the remaining non-wheat pixels are processed based on the Winter Crop Index (WCI) (Yang et al., 2023) and automatic thresholding methods (Otsu algorithm) (Otsu, 1979). Specifically, for the winter growing season (October–June), the remaining non-wheat pixels were classified into winter crops (non-wheat) and non-winter crops using a binary classifier. Similarly, for the spring growing season (April–August), another binary classifier was applied to the remaining non-wheat pixels to separate spring crops (non-wheat) from non-spring crops. Then, non-winter and non-spring crops were classified into forest, water, built-up, and others based on their NDVI characteristics. Taking winter wheat as an example, the general classification process is illustrated in Fig. 7. Following our previous work (Liu et al., 2024a), 500 random points were selected for each class, and spectral separability index (SI) between wheat and five non-wheat land cover types were calculated on a monthly basis. The SI is used to assess the sensitivity of the spectral separability of two classes under certain conditions, determined by the ratio of inter-class and intra-class variability (Somers and Asner, 2013). The higher the value, the better the separation between the two classes in the specified condition. This analysis quantitatively assessed the discriminative power of 25 spectral features (15 vegetation indices and 10 Sentinel-2 spectral bands) at different time periods (Somers and Asner, 2013). A weighted averaging approach was applied to integrate all SI results, producing an overall separability score relative to wheat. To address the potential masking of highly discriminative but unevenly distributed features by mean-based aggregation, a threshold-based filtering mechanism was introduced to exclude features with separability scores below 0.5, thereby enhancing the effectiveness and distinctiveness of feature selection. Finally, for each province, the top five spectral features with the highest mean separability scores were used and combined with Sentinel-1 VV and VH polarization bands to construct a province-specific feature set for wheat mapping.

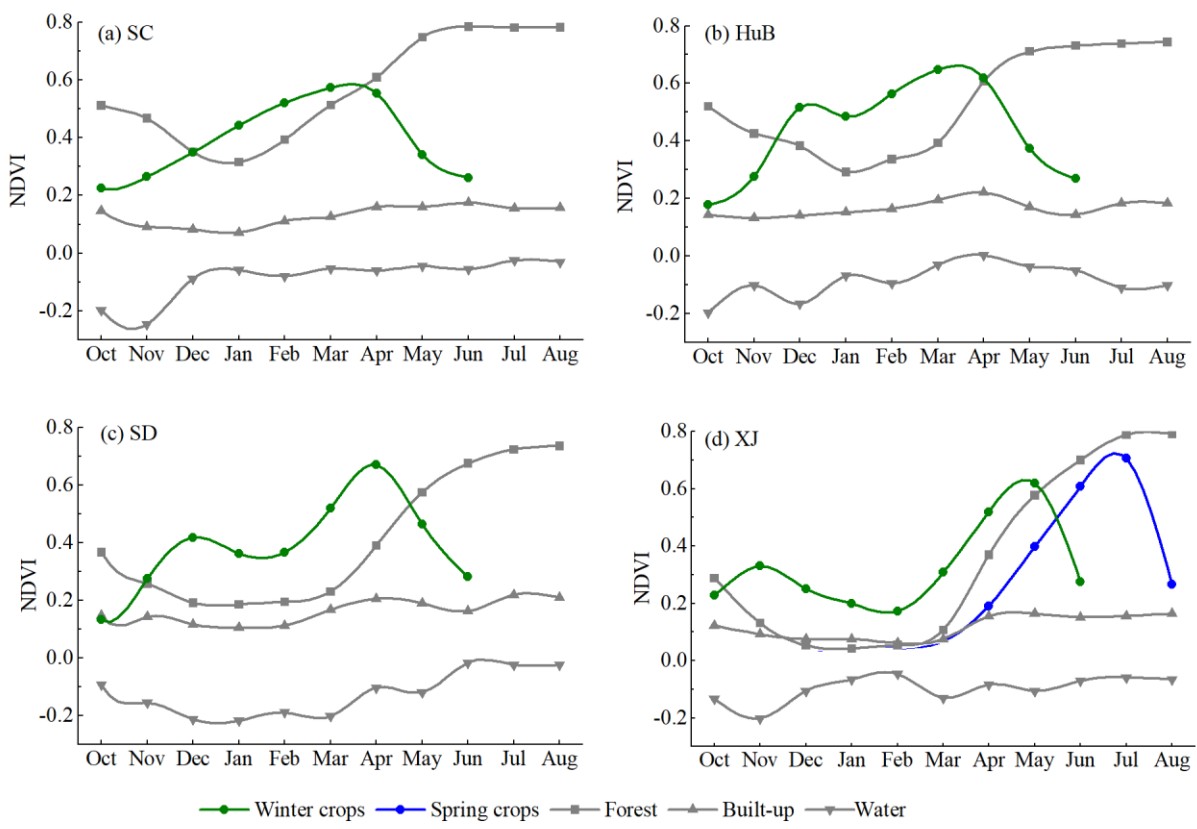

**Figure 6: NDVI curves for different land cover types in four provinces.**

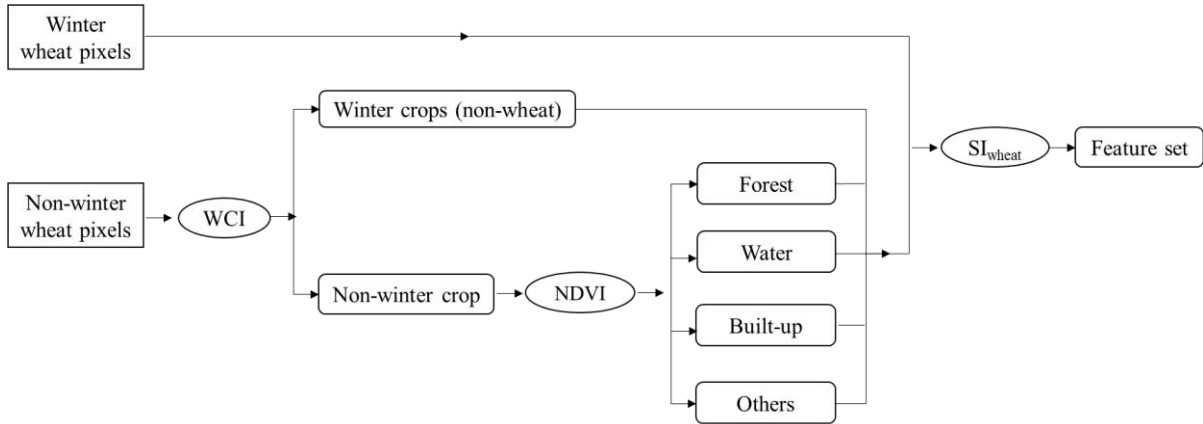

**Figure 7: Flowchart of non-wheat crop classification and wheat feature set selection.**

## 3.3 Mapping and accuracy evaluation of wheat annual distribution

Based on the constructed wheat sample dataset for China and the province-specific remote sensing feature sets, annual wheat distribution maps from 2018 to 2024 were generated using the RF classifier on the GEE. The classifier was implemented with 100 decision trees, there was no more significant difference in accuracy starting with 100 trees and continuing until 200 trees, as shown in Fig. S6. The remaining parameters were maintained at their default values, following the approach adopted in recent remote sensing studies (Yang et al., 2023; Liu et al., 2024b). Given the long growth cycle of winter wheat, there is often a temporal and spatial mismatch between the planted area and the final harvested area. April represents the peak of the wheat growing season, when the canopy is well developed and spectral characteristics are stable and distinct, making it an optimal period for winter wheat identification using remote sensing imagery (Qiu et al., 2017; Dong et al., 2020; Feng et al., 2019; Cai et al., 2018). The middle and late April is the key stage for winter wheat to enter heading. The subsequent grain filling period is easily affected by meteorological disasters such as dry-hot wind, which will cause obvious yield reduction or even no harvest in some areas. The period from early October to early April captures the full early growth stages of winter wheat, including sowing, overwintering, greening, and jointing. Remote sensing imagery acquired during this window is more representative of the actual planted area (Hu et al., 2024). Therefore, to identify the winter wheat planted area, Sentinel-2 imagery from early October to early April was used. To map the winter wheat harvested area, imagery from early October to late June was utilized. For spring wheat, the harvested area was identified based on imagery from early April to late August during the same period. All remote sensing time series were generated at 10-day intervals to ensure faster and more reliable crop type detection. The final products include harvested area maps of spring and winter wheat for 15 provinces, as well as planted area maps of winter wheat for 10 provinces.

It is important to note that the delineation of "planted area" and "harvested area" in this study was not based on independent labels explicitly recording planting or harvesting events, but rather on adjusted temporal windows designed to capture key phenological phases of wheat growth. The maps derived from temporal window adjustment can be interpreted as phenology-based representations of winter wheat distribution. Specifically, the "planted area map" is phenologically closer to an in-season distribution, while the "harvested area map" is more comparable to an end-season distribution. Nevertheless, this correspondence should be regarded as an interpretive perspective rather than a strict equivalence to single-date mid-season or end-season classification results. In addition, to satisfy the logical requirement that the harvested area should be a subset of the planted area, the harvested area in this study was masked within the extent of the planted area.

Three complementary data sources were integrated to assess product accuracy and stability. First, large-scale field survey and manually labelled validation samples covering 15 provinces were used to calculate standard mapping accuracy metrics, including Overall Accuracy (OA), User's Accuracy (UA), and Producer's Accuracy (PA) (Liu et al., 2024a). Second, spatial consistency comparisons were conducted with existing publicly available remote sensing-based wheat maps to assess the spatial distribution reliability of the CN_Wheat10 product. Third, a quantitative regression analysis was performed using

provincial- and municipal-level wheat area statistics from the China Statistical Yearbook. The R² was used as the evaluation
metric to quantify the product's area-based accuracy across different administrative levels (Liu et al., 2024b).

## 4 Results

### 4.1 Comparison with existing wheat maps

Figure 8 presents the spatial distribution of spring-winter wheat across China, delineating the nationwide patterns of both crop types. To enhance the understanding of spatial details, 18 representative regions were selected for zoomed-in visualization.
The planted area (CN_Wheat10(P)) and harvested area (CN_Wheat10(H)) were compared with existing publicly available remote sensing products. Spring wheat is predominantly distributed in northwest China, including five provinces: XJ, GS, NX, QH, and NM. In Sites 1–4, we compared the CN_Wheat10 with the WorldCereal spring cereal map in 2021. The results showed that the identified wheat areas were largely consistent between the two products and exhibited high spatial agreement with wheat-growing regions observed in Sentinel-2 imagery. In some regions, CN_Wheat10 delineated spring wheat fields
more precisely, with clearer representation of field boundaries and roads. In Sites 5–8, we compared the CN_Wheat10 with the ChinaCP-Wheat10m spring wheat map in 2020. The ChinaCP-Wheat10m results exhibited excessive noise, blurred field boundaries, and poor spatial continuity, whereas CN_Wheat10 demonstrated superior classification performance and spatial consistency, particularly in clearly distinguishing spring wheat from bare land and non-cropland. Winter wheat covers a much broader region, mainly concentrated in eastern China's Huang-Huai-Hai region, including the provinces of HN, SD and HB
provinces. When compared to existing remote sensing products, CN_Wheat10 demonstrates significant advantages in identifying winter wheat. It not only achieves higher mapping accuracy but also maintains complete spatial coverage. For example, in Site 14 (Jining, SD province), the dark green areas in the Sentinel-2 imagery represent winter wheat, while light green areas are predominantly garlic fields. Several existing products show notable misclassification in this region, incorrectly identifying garlic as wheat and thereby reducing mapping precision. In contrast, CN_Wheat10 effectively distinguishes
between the two crops, accurately excluding interference from non-wheat vegetation.

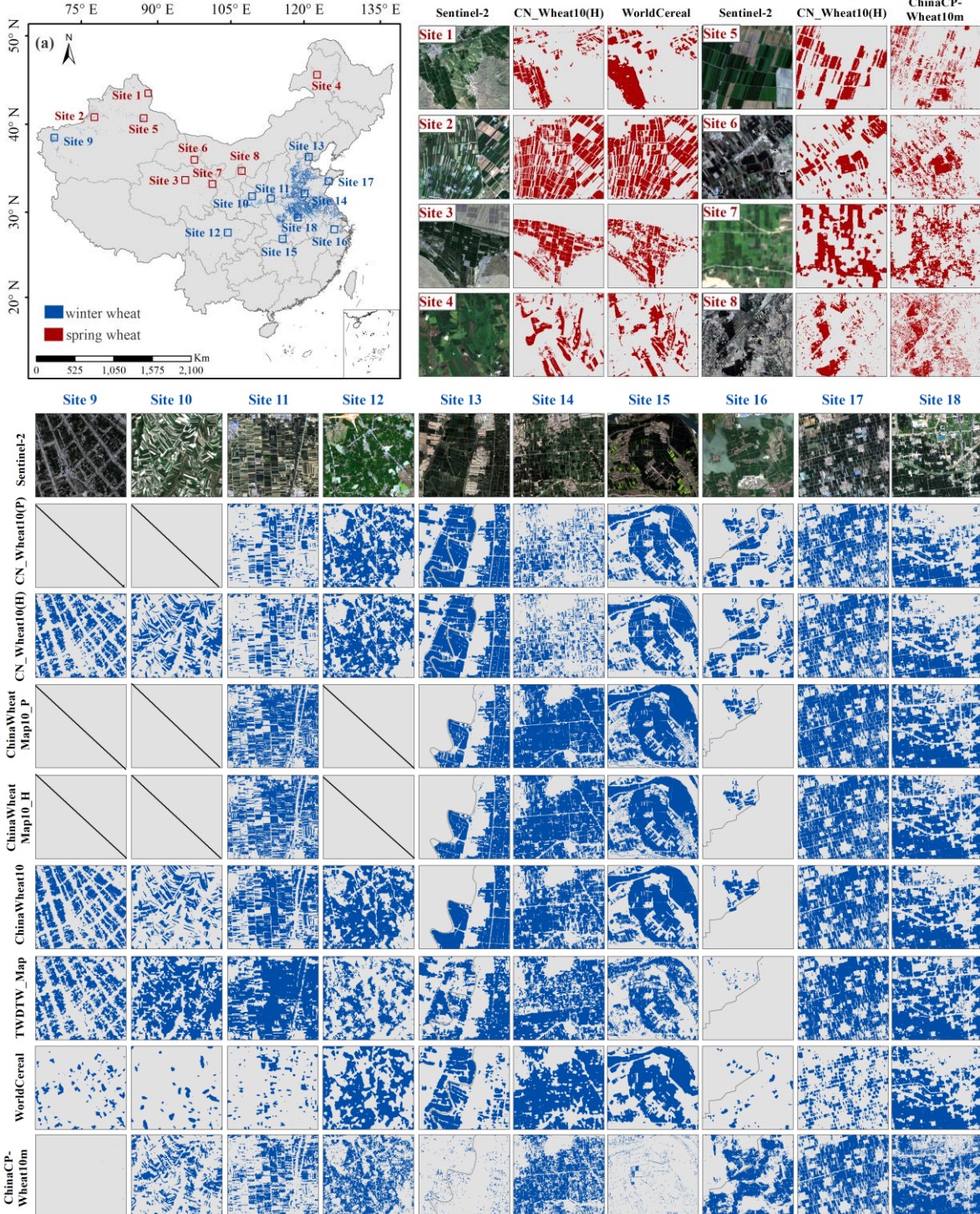

**Figure 8: Comparison of wheat details between CN_Wheat10 products and existing published products.**

Figure 9 systematically summarizes the mapping accuracy for spring and winter wheat from 2018 to 2024. Across multiple accuracy metrics, the CN_Wheat10 product demonstrates notable advantages and stable performance in mapping the spatial distribution. Specifically, for winter wheat, the planted area accuracy (CN_Wheat10(P)) consistently exceeds 0.96, while the harvested area accuracy (CN_Wheat10(H)) remains above 0.95, significantly outperforming existing comparable products. For spring wheat, although the mapping accuracy shows slight fluctuations (ranging from 0.919 to 0.987) due to ecological heterogeneity and the complexity of crop types in its growing regions, the overall accuracy remains at a high level. Taken together, CN_Wheat10 exhibits strong temporal-spatial reliability, with high interannual consistency and robust mapping performance.

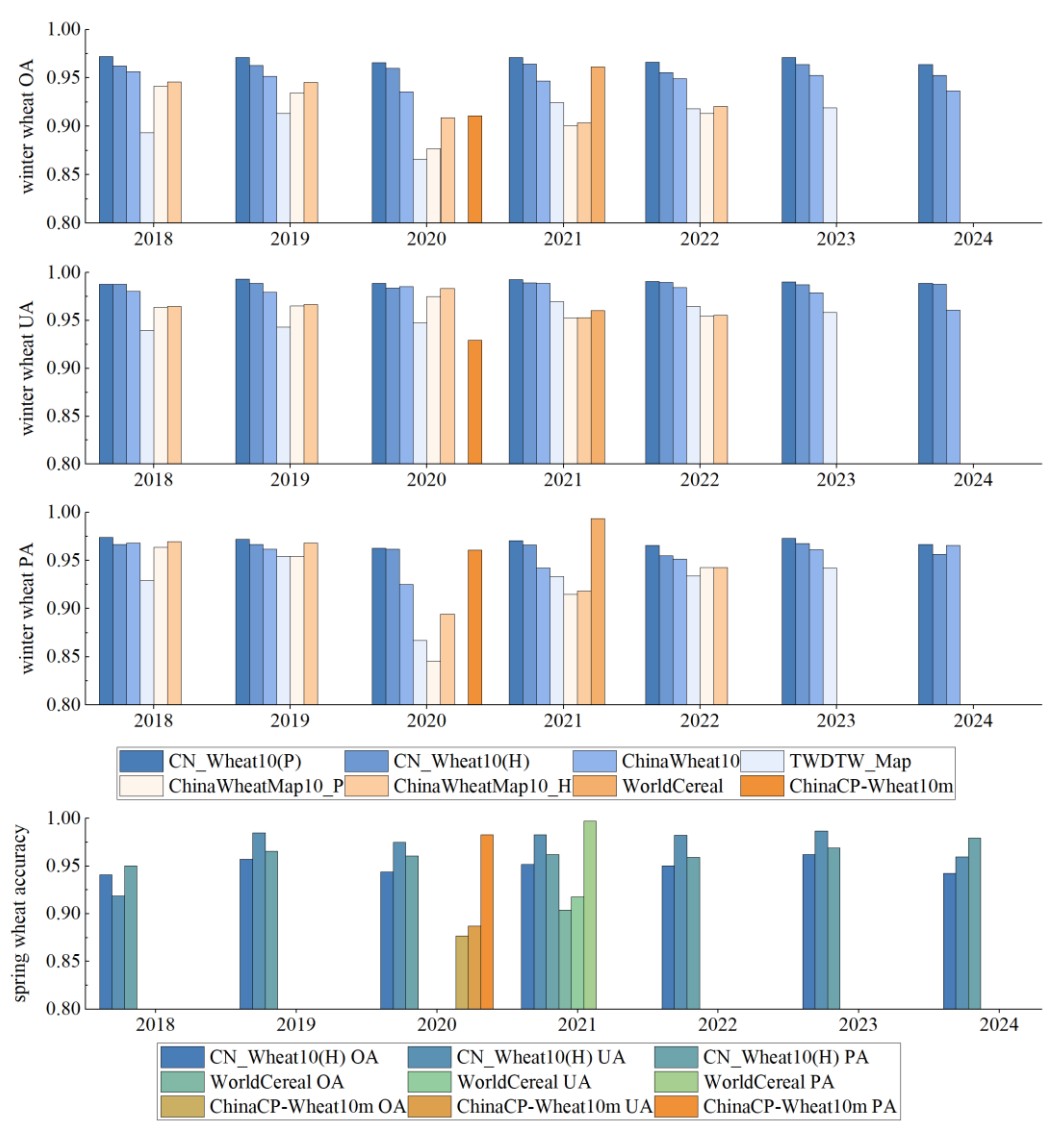

**Figure 9: The mapping accuracy for spring and winter wheat from 2018 to 2024.**

We further analyzed the mapping accuracy of wheat at the provincial level. As shown in Fig. 10, the CN_Wheat10 product demonstrates consistently high accuracy across all provinces, with particularly outstanding performance in regions dominated by a single wheat type. For instance, in major winter wheat production areas such as the Huang-Huai-Hai region, where cropping structures are stable and phenological stages are well synchronized, the average planting accuracy exceeds 0.95, and the average harvesting accuracy surpasses 0.94. In northwest provinces such as XJ, GS, and NX, where both spring and winter wheat coexist and their phenological cycles partially overlap, spectral confusion remains a challenge in certain years and regions. Nonetheless, CN_Wheat10 maintains high mapping accuracy even under these complex agro-ecological conditions. The Fig. 10 shows that ChinaCP-Wheat10m and WorldCereal achieve relatively high accuracies in certain provinces. This is largely because both products provide wheat distribution maps for a single year, reflecting the accuracy for that specific year only, whereas the other products report multi-year average accuracies. Notably, the accuracy of mapping planted areas is generally higher than during the harvested area. This discrepancy can be attributed not only to the inherent spectral differences in the remote sensing time series but also to the influence of natural hazards during wheat development. During the harvest stage, some wheat fields may be affected by extreme weather events such as hot-dry winds, floods, or pest and disease outbreaks, which can lead to premature senescence, yield loss, or even crop failure. These stress-induced changes often result in sharp declines or irregular fluctuations in vegetation indices, weakening the expression of typical wheat spectral patterns and increasing the likelihood of misclassification or confusion in remote sensing-based harvest-stage mapping.

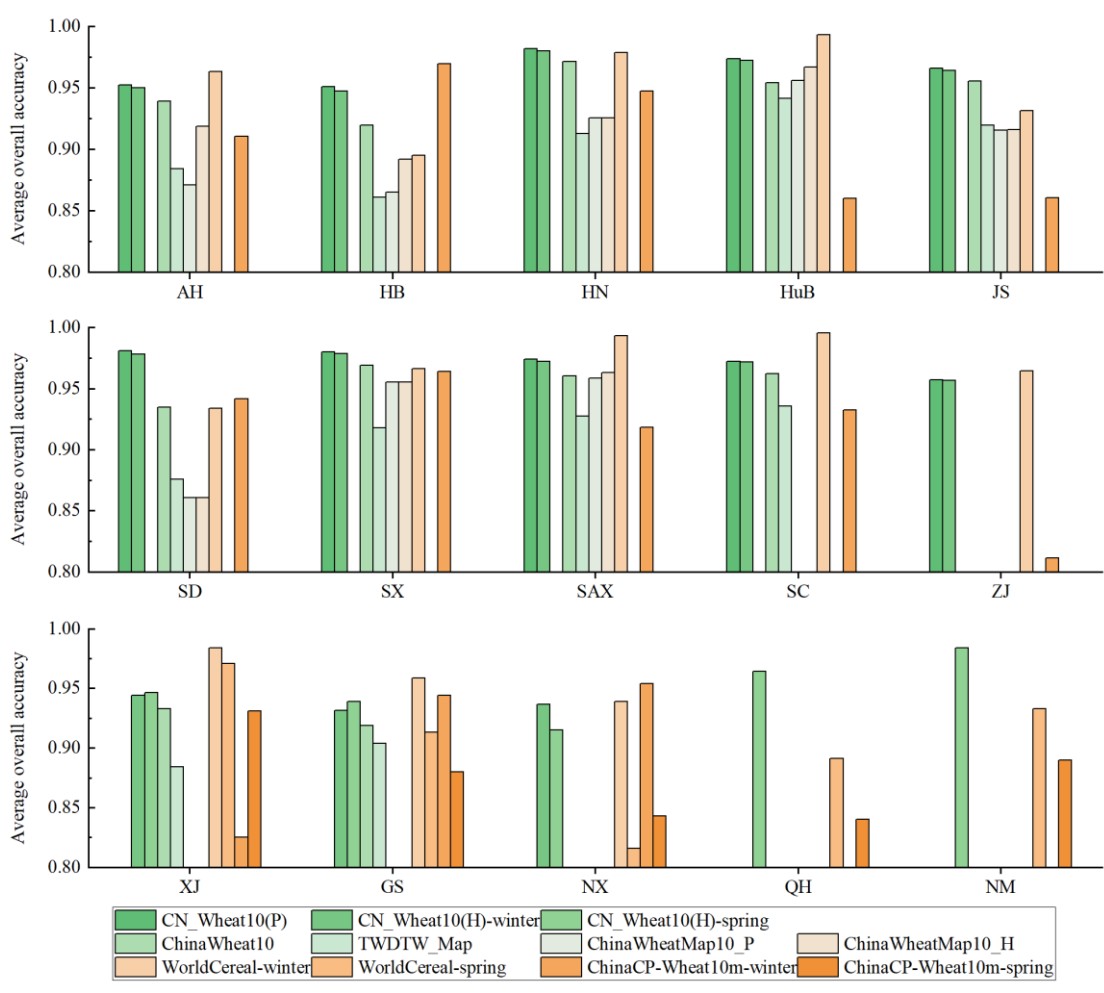


**Figure 10: The average overall accuracy of wheat at the provincial level from 2018 to 2024.**

### 4.2 Comparison with agricultural statistics

The annual wheat areas and their associated uncertainties for the period 2018–2024 were estimated using the method recommended by existing studies (Olofsson et al., 2014), with key results summarized in Table 2. For the ten major wheat

provinces, the estimated planted area ranged from 16,680.34 ± 431.36 thousand hectares to 19,702.84 ± 277.17 thousand hectares, accounting for an average of approximately 10.6% of the total area. The estimated harvested area in these provinces ranged from 16,121.57 ± 561.61 thousand hectares to 19,130.85 ± 407.08 thousand hectares, averaging about 10.14% of the total. For the fifteen wheat provinces, the estimated harvested area was between 20,362.92 ± 1,513.43 and 24,684.57 ± 2,049.75 thousand hectares, with a mean proportion of approximately 3.91%. Overall, the annual wheat maps demonstrate good

performance, as reflected in the 95% confidence intervals of the area estimates. It is important to note that the relatively higher uncertainty in the harvested area estimates is primarily attributed to the class imbalance in the stratified validation samples.

This does not necessarily indicate poor map accuracy but reflects a statistical limitation of the sampling design, a phenomenon documented in previous studies (Liu and Zhang, 2023; Yadav and Congalton, 2017).

**Table 2: Error-adjusted area estimates for annual CN_Wheat10 from 2018 to 2024.**

| Year | Attribute/Strata | planted area (10 provinces) | harvested area (10 provinces) | harvested area (15 provinces) |
|------|------------------|------------------------------|-------------------------------|-------------------------------|
| 2018 | Area proportion (%) | 10.18 | 9.58 | 3.86 |
|      | Standard error ($\times 10^3$ ha) | 244.24 | 270.53 | 752.89 |
|      | Estimated area ($\times 10^3$ ha) | 17607.72 | 16570.03 | 21980.48 |
|      | 95% CI in $\pm$ ($\times 10^3$ ha) | 478.71 | 530.25 | 1475.67 |
| 2019 | Area proportion (%) | 9.64 | 9.32 | 3.57 |
|      | Standard error ($\times 10^3$ ha) | 220.08 | 286.54 | 772.16 |
|      | Estimated area ($\times 10^3$ ha) | 16680.34 | 16121.57 | 20362.92 |
|      | 95% CI in $\pm$ ($\times 10^3$ ha) | 431.36 | 561.61 | 1513.43 |
| 2020 | Area proportion (%) | 10.16 | 9.81 | 3.96 |
|      | Standard error ($\times 10^3$ ha) | 155.55 | 223.10 | 715.43 |
|      | Estimated area ($\times 10^3$ ha) | 17583.70 | 16966.48 | 22552.06 |
|      | 95% CI in $\pm$ ($\times 10^3$ ha) | 304.88 | 437.27 | 1402.24 |
| 2021 | Area proportion (%) | 11.39 | 11.06 | 4.15 |
|      | Standard error ($\times 10^3$ ha) | 141.41 | 207.69 | 700.33 |
|      | Estimated area ($\times 10^3$ ha) | 19702.84 | 19130.85 | 23654.27 |
|      | 95% CI in $\pm$ ($\times 10^3$ ha) | 277.17 | 407.08 | 1372.66 |
| 2022 | Area proportion (%) | 10.73 | 10.26 | 3.84 |
|      | Standard error ($\times 10^3$ ha) | 162.02 | 227.36 | 745.15 |
|      | Estimated area ($\times 10^3$ ha) | 18560.11 | 17752.06 | 21863.52 |
|      | 95% CI in $\pm$ ($\times 10^3$ ha) | 317.56 | 445.63 | 1460.50 |
| 2023 | Area proportion (%) | 11.09 | 10.46 | 4.33 |
|      | Standard error ($\times 10^3$ ha) | 317.15 | 369.83 | 1045.79 |
|      | Estimated area ($\times 10^3$ ha) | 19181.10 | 18096.78 | 24684.57 |
|      | 95% CI in $\pm$ ($\times 10^3$ ha) | 621.62 | 724.86 | 2049.75 |
| 2024 | Area proportion (%) | 11.01 | 10.53 | 3.69 |
|      | Standard error ($\times 10^3$ ha) | 269.37 | 289.78 | 650.31 |
|      | Estimated area ($\times 10^3$ ha) | 19042.65 | 18219.26 | 21003.42 |
|      | 95% CI in $\pm$ ($\times 10^3$ ha) | 527.97 | 567.97 | 1274.61 |


To assess the applicability of the CN_Wheat10 product in estimating wheat areas, we conducted a systematic comparison between the planting and harvesting areas and the official agricultural statistics of China from 2018 to 2023 (Fig. 11–14). In this study, the areas of two wheat types were combined and analyzed. The results show a high level of consistency between CN_Wheat10 estimates and the official statistics across multiple spatial scales, indicating strong agreement. Specifically, the

$R^2$ for provincial-level planted area ranges from 0.948 to 0.979, while for the municipal level it ranges from 0.892 to 0.934. For harvested area, the $R^2$ for provincial-level values range from 0.951 to 0.976, and from 0.889 to 0.926 at the municipal level. These findings demonstrate that the CN_Wheat10 product not only effectively captures the overall spatial patterns of wheat cultivation at a national scale, but also their ability to capture spatial detail, which is suitable for more sophisticated agricultural management and policy formulation needs.

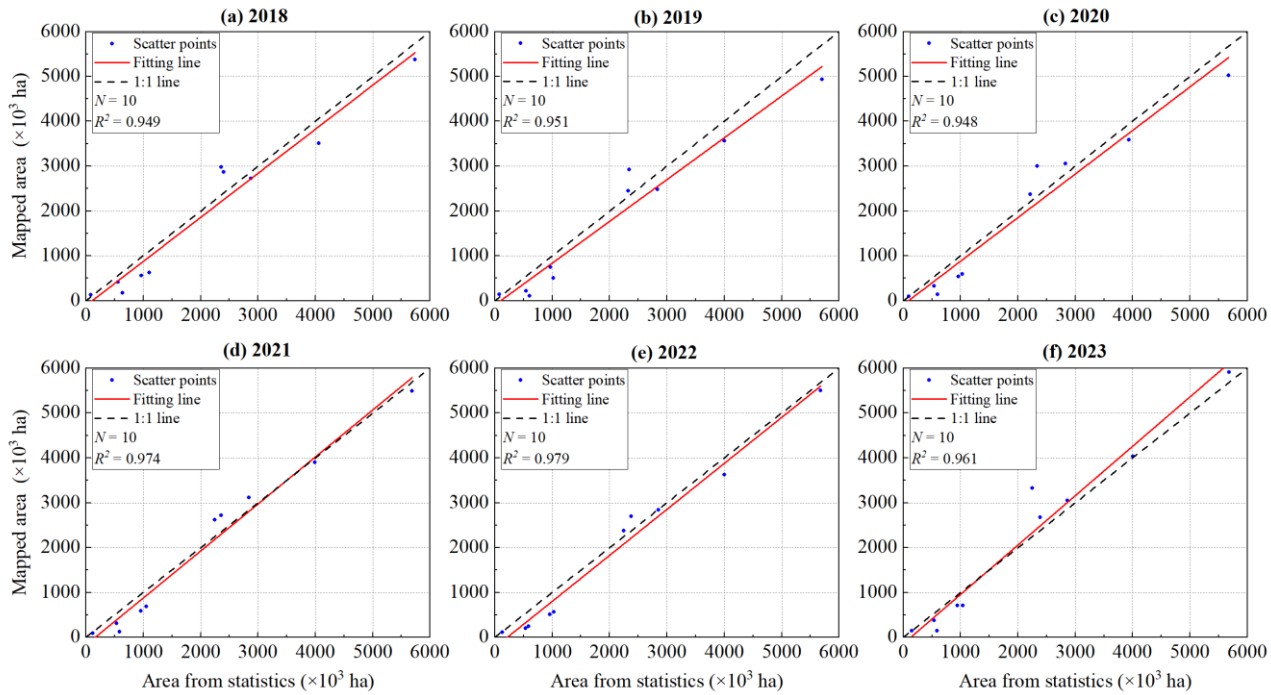


**Figure 11: Provincial comparison of wheat planted area (CN_Wheat10(P)) with Statistical Yearbook data.**

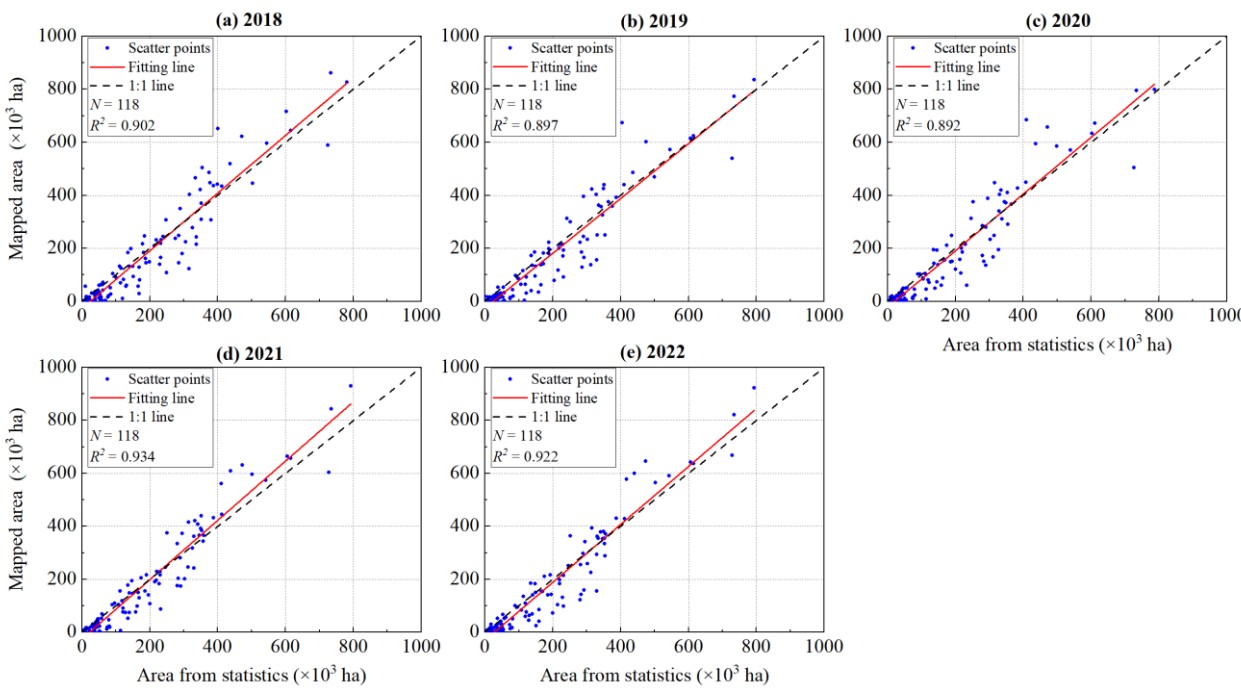

**Figure 12: Municipal comparison of wheat planted area (CN_Wheat10(P)) with Statistical Yearbook data.**


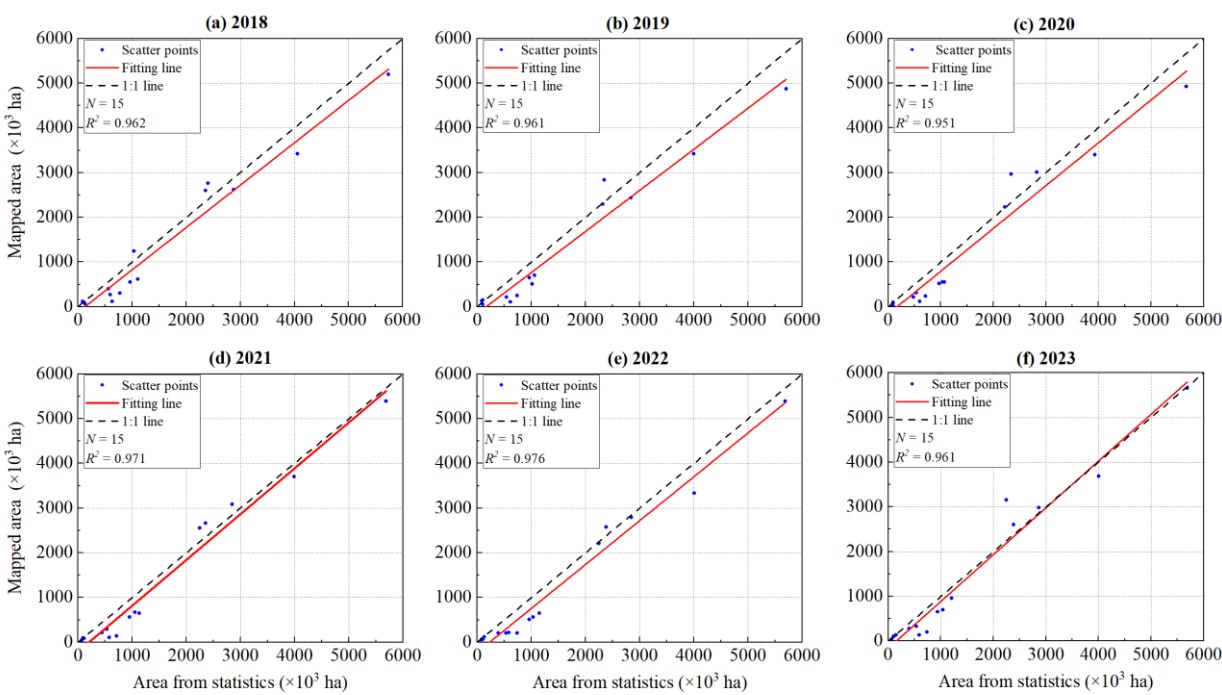

**Figure 13: Provincial comparison of wheat harvested area (CN_Wheat10(H)) with Statistical Yearbook data.**

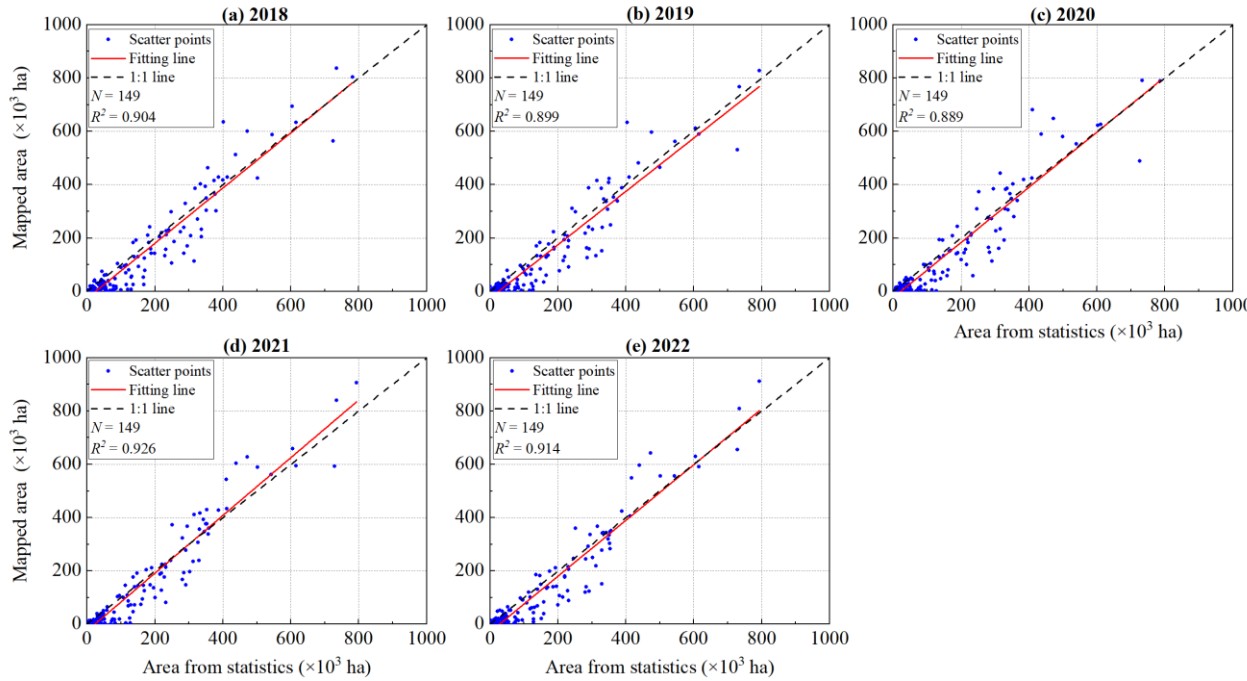

**Figure 14: Municipal comparison of wheat harvested area (CN_Wheat10(H)) with Statistical Yearbook data.**

### 4.3 Discrepancies between winter wheat planted and harvested areas

Figure 15 illustrates the spatial differences between wheat planted and harvested area across major wheat-producing regions in China. Overall, some inconsistencies were observed between the two map types, particularly in the provinces of SD, HN, and HB, which represent the core winter wheat production zones with the largest cultivation areas and highest sowing densities in China. To further quantify these spatial discrepancies and analyze their temporal trends, we conducted a statistical analysis of the annual area difference (i.e., planted area minus harvested area) in 10 provinces during 2018–2024 (Fig. 16). The results

show that the most significant area differences occurred in 2018 and 2023, each exceeding one million hectares, which corresponds to approximately 5% of the total planted area for those years. Spatially, HB, HN, and SD provinces experienced the greatest reductions. In these regions, a considerable proportion of areas identified as wheat in spring could no longer maintain consistent spectral characteristics in summer. It should be emphasized that these "planted–harvested differences" do not represent precise yield losses, but rather provide an indicative and uncertainty-prone measure to reveal the potential

spatiotemporal patterns and relative magnitude of wheat reduction. The observed discrepancies and interannual fluctuations highlight the sensitivity of wheat cultivation to climatic variability and natural hazards, but should be interpreted primarily as qualitative or semi-quantitative signals rather than absolute production loss estimates. Since both the planted and harvested area estimates are derived from statistically based area-estimation methods recommended in previous studies, each estimate is associated with its own standard error. Accordingly, the error of the area difference should be regarded as a combination of

the uncertainties from the two area estimates, and its magnitude can be quantified using standard error-propagation formulas.

Based on the calculations using Table 2, the annual uncertainty range of the area differences across the ten provinces is approximately 251.26 to 487.19 thousand hectares. In addition, the imbalance in class proportions among the validation samples may further increase the uncertainty of area estimation, thereby affecting the stability of the area differences.

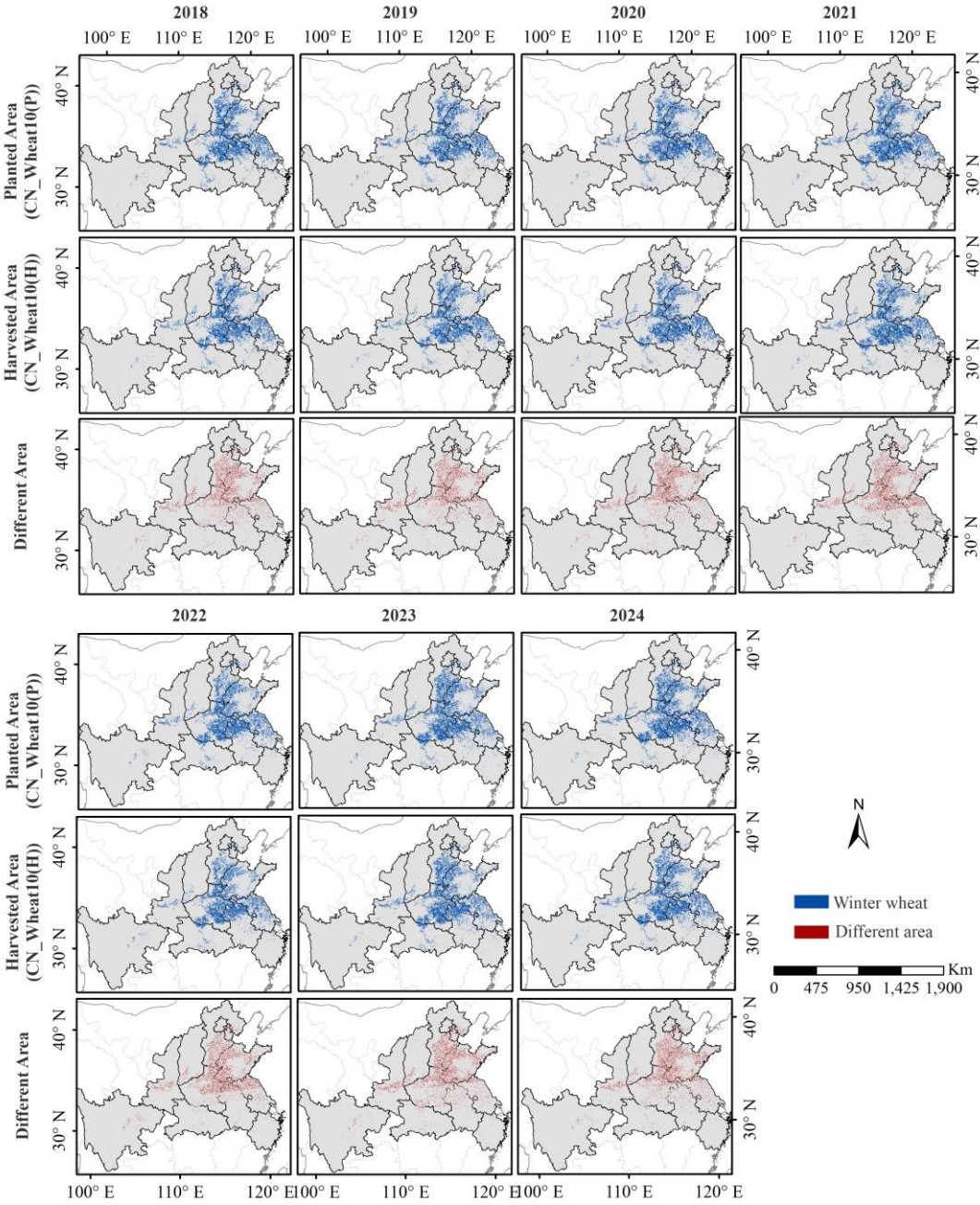

**Figure 15: Comparison of wheat planted and harvested area in 10 provinces.**

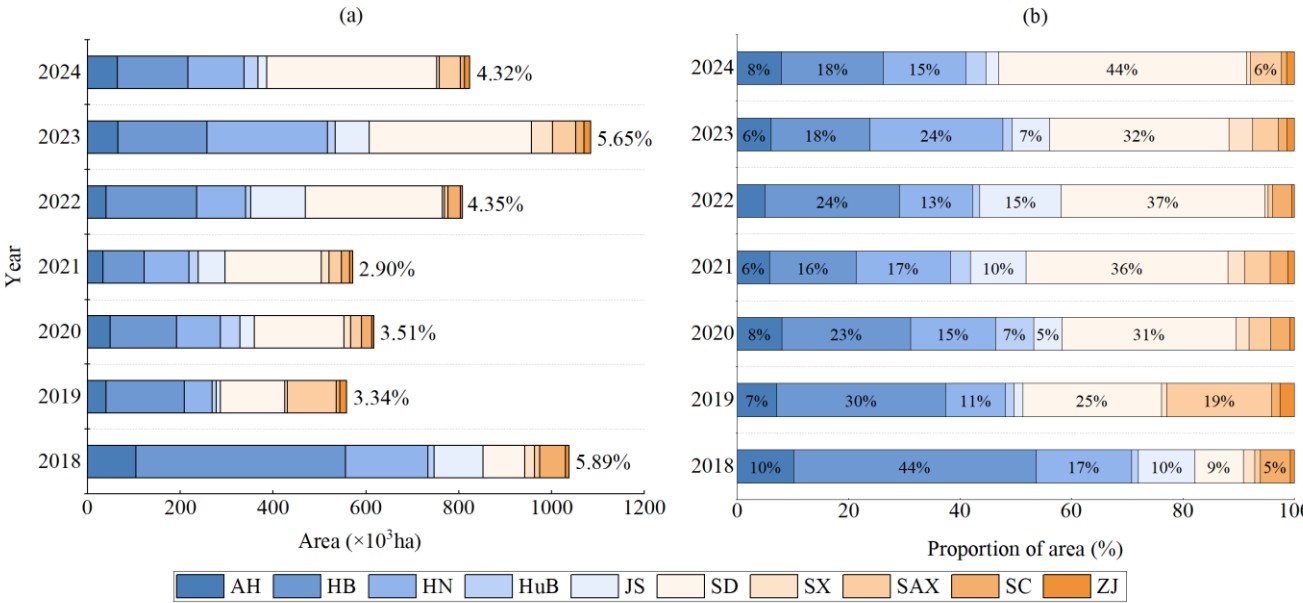

**Figure 16: Differences between planted and harvested wheat area by province from 2018 to 2024. (a) Annual difference between planted and harvested wheat areas (in hectares) and its proportion of the total planted area; (b) Annual percentage of planted–harvested area difference for each province.**


To provide a more intuitive representation of these spatial discrepancies, five representative regions were selected for detailed visualization (Fig. 17). In these regions, the spring-stage remote sensing imagery (typically in April) exhibited characteristic wheat canopy features, such as high vegetation index values and strong reflectance in the green spectral bands, indicating that the wheat was in a vigorous growth phase (typically from stem elongation to early grain filling) with high leaf area index and dense ground cover, making crop identification relatively accurate during this period. However, by mid to late May, a noticeable reduction in wheat extent was observed in some areas during the pre-harvest stage. This change can primarily be attributed to a variety of adverse meteorological and biological factors, including drought stress, hot-dry winds, pest and disease outbreaks, and flooding. These factors may have led to premature senescence, yield reduction, or even total crop failure in certain fields. Such abnormal growth events result in significant spectral changes in remote sensing imagery, where previously vegetated areas with high reflectance become spectrally similar to bare soil or non-crop surfaces, thereby increasing the likelihood of misclassification or exclusion in harvest-stage mapping. It is important to emphasize that the observed "planted area > harvested area" discrepancy does not result solely from remote sensing misclassification, but reflects potential agronomic instability and environmental stress. At the same time, this difference should be interpreted as an indicative, uncertainty-prone measure, used to reveal the potential spatiotemporal patterns and relative magnitude of wheat reduction, rather than as a direct estimate of actual yield loss. By explicitly capturing and analyzing these differences between planting

and harvesting stages, the CN_Wheat10 product provides valuable information on abnormal crop dynamics, supporting applications such as disaster impact assessment, crop insurance verification, and agricultural policy development.

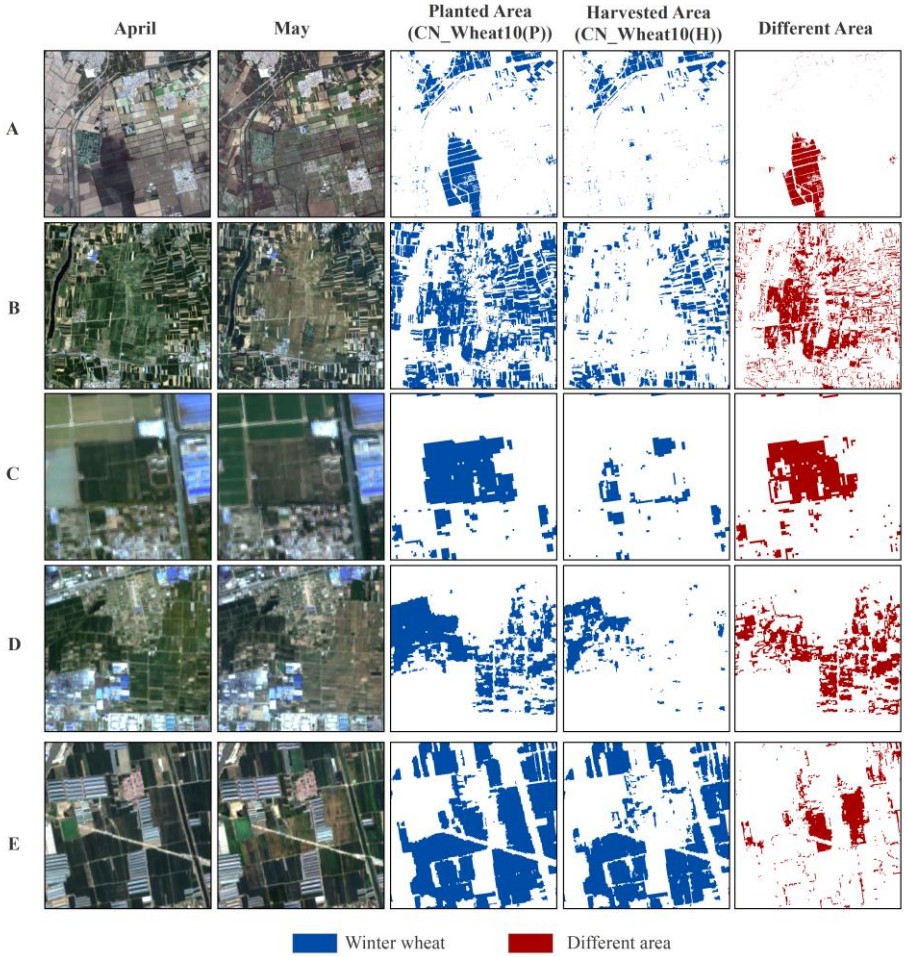

**Figure 17: Spectral characteristics of wheat at different growth stages and differences between planted and harvested area.**

### 4.4 Spatial distribution pattern of wheat in China

As depicted in Fig. 18, the distribution of wheat cultivation in China exhibits a distinct pattern characterized by "concentration in the east, dispersion in the west," with winter wheat dominating and spring wheat serving a supplementary role. At the regional scale, wheat planting shows marked spatial heterogeneity. In eastern China, the Huang-Huai-Hai region represents the primary production zone for winter wheat. This region features flat terrain, fertile soils, and well-developed irrigation infrastructure. Moreover, its favorable climatic conditions create an optimal environment for winter wheat to overwinter safely and achieve stable, high yields. Consequently, large-scale, contiguous, and highly intensive winter wheat cultivation has been established in this region, making it the core area with the highest planting area of winter wheat. In contrast, the central hilly

regions are constrained by rugged topography and fragmented arable land. Here, wheat cultivation exhibits a pronounced terraced pattern. Although some areas maintain winter wheat at moderate scales, the lack of large contiguous fields, combined

with lower levels of mechanization and farm management, limits the overall planting scale. In northwest China, spring wheat is predominant. However, its spatial distribution is relatively scattered and typified by an "oasis agriculture" pattern. These areas are generally arid, with low precipitation, and agricultural development is highly dependent on irrigation. Major wheat-producing zones are primarily located in irrigated oases along the edges of the Tarim Basin, the Hexi Corridor, and the Hetao Plain.

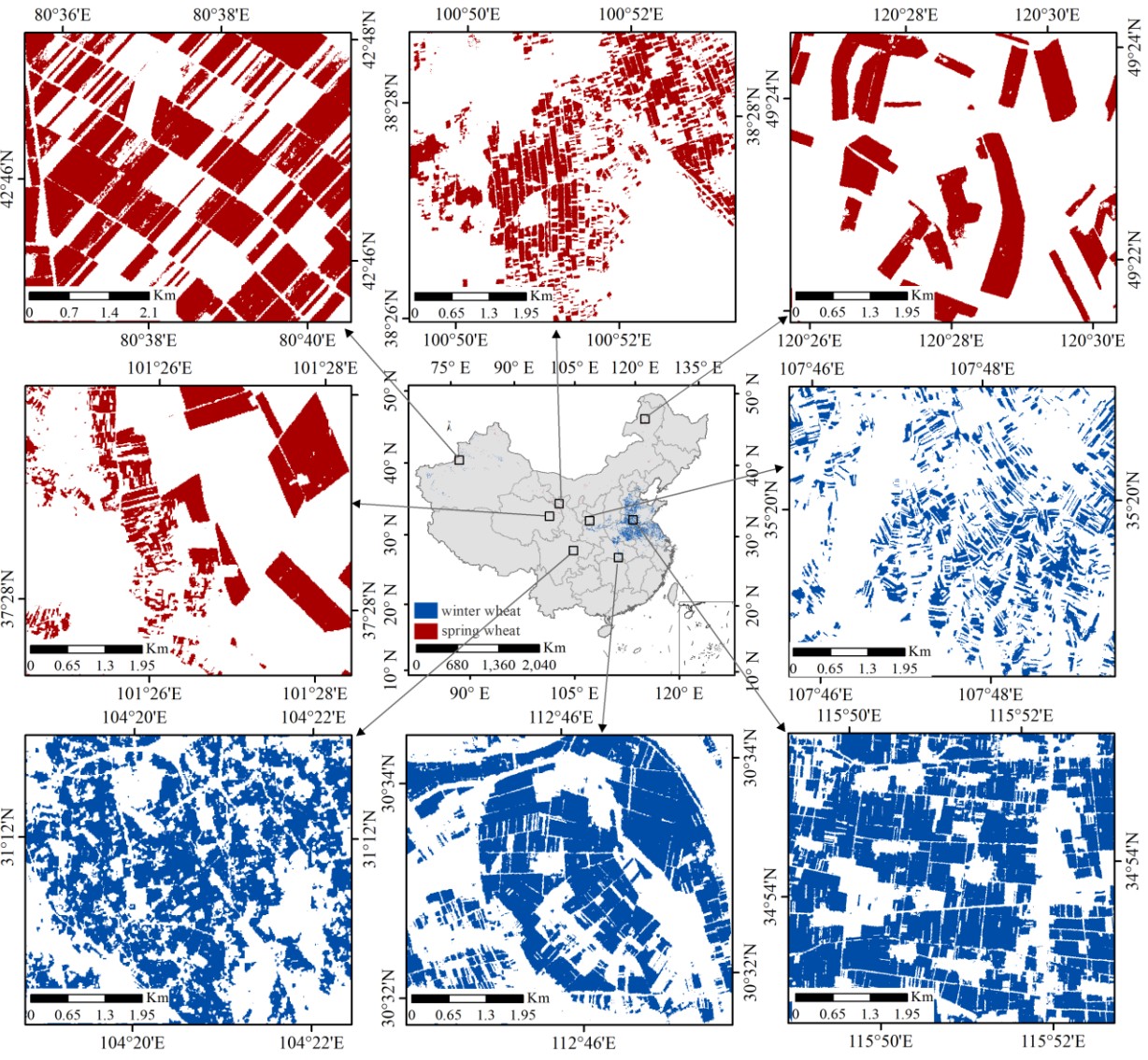

**Figure 18: Distribution pattern of spring and winter wheat in China.**

## 5 Discussions

Based on a systematic wheat sample generation strategy and a province-level feature selection approach, we developed a high spatiotemporal resolution dataset of spring and winter wheat distributions (CN_Wheat10), which effectively fills the existing gap in spring wheat monitoring. CN_Wheat10 dataset covers the harvested areas of spring-winter wheat across 15 provinces in China, as well as the planted areas of winter wheat in 10 provinces, spanning the period from 2018 to 2024. CN_Wheat10 provides a robust data foundation for applications such as food security monitoring, agricultural management, and crop growth modelling.

### 5.1 Advantages of the CN_Wheat10 dataset

The systematic sample generation strategy adopted in this study ensures the representativeness and mapping accuracy of the CN_Wheat10 dataset. While numerous automated sample generation approaches have been proposed in recent research, many of these methods tend to treat winter rapeseed as the primary confusion class in winter wheat identification, while overlooking crops such as garlic that share highly similar phenological characteristics with winter wheat (Fu et al., 2025; Yang et al., 2024; Dong et al., 2020). This limitation is particularly problematic in regions with widespread mixed cropping or crop rotation, where sample purity may be compromised, ultimately reducing the mapping model's performance and generalizability. To address this issue, we propose a cross-regional sample generation method that integrates time-series remote sensing imagery with existing crop distribution products. This approach leverages the phenological dynamics captured in temporal satellite data and incorporates geographic knowledge and regional cropping structure to enforce multi-dimensional constraints during sample selection. This strategy not only enhances the inter-class separability of samples but also significantly improves their spatiotemporal diversity and consistency. Especially in the main spring wheat producing areas, due to the difficulty of sample acquisition and strong spatial and temporal heterogeneity, historical research has obvious shortcomings in sample construction. Moreover, the cross-regional sample generation strategy based on existing products proves to be practical and replicable in real-world applications, greatly minimizing dependence on comprehensive field surveys for data sampling (Li et al., 2024; Tran et al., 2022). By more effectively excluding highly confounding crops such as garlic, the method also increases the class purity of wheat in remote sensing mapping, providing technical support for the development of stable and high-precision spring and winter wheat distribution products.

The feature selection process at the provincial scale significantly enhanced the regional adaptability of the mapping model. Given China's vast geographic expanse, the wide distribution of wheat-growing regions, and substantial regional variation in phenological characteristics (Tao et al., 2012), a unified set of feature variables often fails to meet the crop identification requirements across all areas. To address this limitation, we implemented a differentiated feature selection strategy at the provincial level. This approach adapts the input variable combinations based on each province's wheat phenological development, cropping structure, and characteristics of potential confusion crops, thereby allowing the model to better capture local wheat growth patterns and temporal dynamics. Taking Henan Province as an example, we calculated the variation of

wheat precision before and after feature selection (Table 3). Without provincial feature selection, the overall precision ranged

from 0.960 to 0.980; after feature selection, the overall precision increased to 0.974 to 0.988. Although the model already achieved relatively high accuracy without feature selection, the use of provincial feature selection further enhanced its discriminative capacity. To better illustrate this improvement, six representative locations were selected for comparison in Fig. 19, where it is evident that feature selection enabled more precise spatial identification of winter wheat, thereby increasing the reliability and robustness of the results. This region-specific strategy mitigates the risk of generalization failure commonly

observed in "one-size-fits-all" models when applied across heterogeneous regions. It thus provides a scalable and widely applicable framework for remote sensing-based crop mapping. According to the statistical results presented in Table S3, the top five most frequently selected spectral variables across provinces highlight notable regional differences in the importance of crop identification features. As shown in Fig. S7, the high selection frequency of the Normalized Red-edge3 Difference Vegetation Index (NREDI3) and Normalized Red-edge2 Difference Vegetation Index (NREDI2) underscores the critical role

of red-edge bands in wheat mapping. These indices are particularly effective in distinguishing different growth stages and reflecting crop health status (Delegido et al., 2013; Qiu et al., 2025). The vegetation vigor indices such as the Optimized Soil Adjusted Vegetation Index (OSAVI) and NDVI remain core indicators of wheat identification performance, reflecting the fundamental importance of plant growth conditions (Qu et al., 2021; Zhao et al., 2020; Radočaj et al., 2023). Notably, in provinces with significant winter rapeseed cultivation, spectral indices such as the Normalized Difference Yellowness Index

(NDYI) and the Winter Rapeseed Index (WRI) were found to play a substantial role in model performance (Zhang et al., 2022a; Sulik and Long, 2016). It can be inferred that the province-specific feature selection approach not only improves wheat mapping accuracy but also strengthens the model's ability to distinguish wheat from spectrally similar crops.

**Table 3: Comparison of wheat recognition accuracy in Henan Province before and after feature selection.**

|  | 2018 | 2019 | 2020 | 2021 | 2022 | 2023 |
|---|---|---|---|---|---|---|
| No feature selection | 0.976 | 0.974 | 0.979 | 0.980 | 0.964 | 0.960 |
| Feature selection | 0.987 | 0.979 | 0.982 | 0.988 | 0.982 | 0.974 |

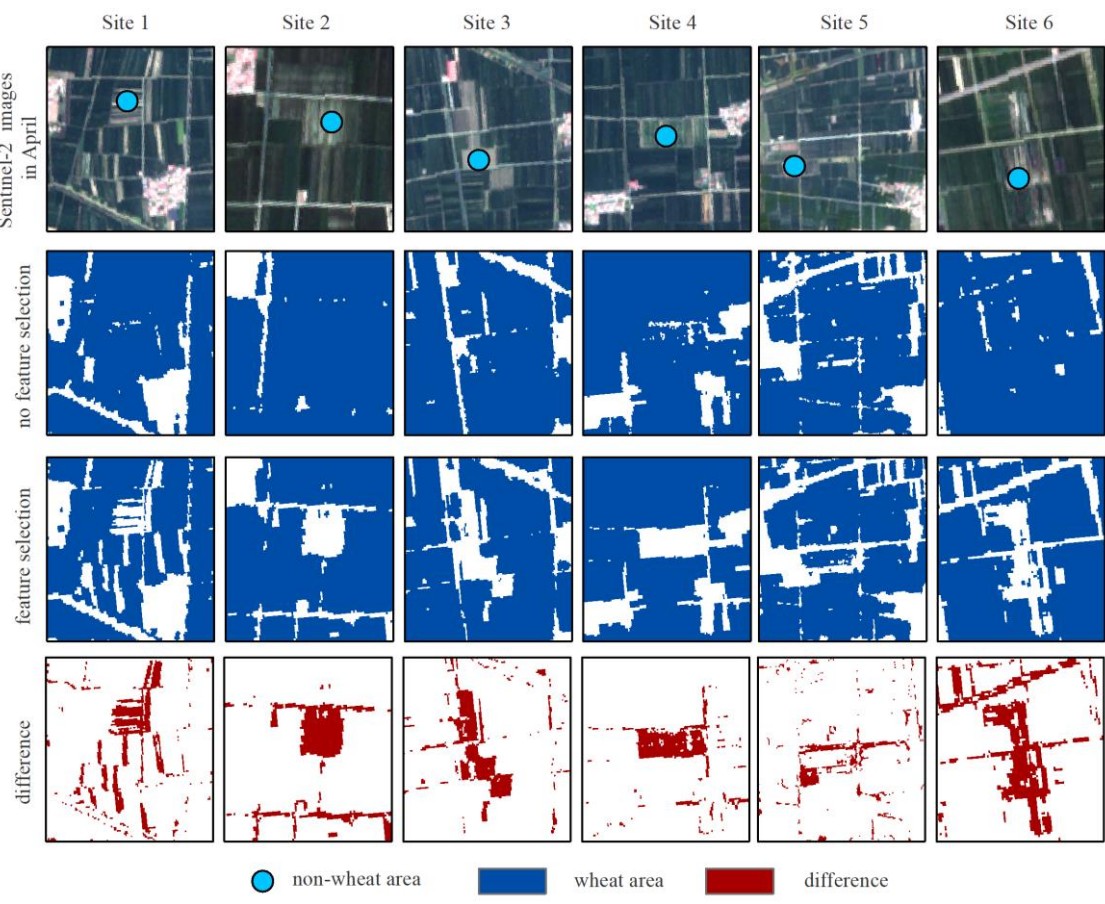

   **Figure 19: Comparison of wheat remote sensing recognition regions before and after feature selection.**

## 5.2 Uncertainties of the CN_Wheat10 dataset

The "planted area" and "harvested area" maps generated in this study are essentially derived from remote sensing observations corresponding to different phenological stages. Specifically, the "planted area" represents wheat distribution identified from imagery covering the overwintering to regreening period, reflecting fields that successfully established and survived early growth, and can be interpreted as the "potential planted area." The "harvested area" is further derived by incorporating spectral information from heading to maturity on top of the planted area, aiming to capture fields that successfully completed key reproductive growth and reached a harvestable state, approximating the "actual harvested area." The choice of the terms "planted" and "harvested" is intended to more intuitively convey their agronomic relevance and maintain consistency with previous studies (Hu et al., 2024). Nevertheless, it should be emphasized that these maps can also be interpreted, in a strict sense, as "in-season" and "end-season" distributions. The differences in area between the two maps largely reflect dynamic changes in crop extent caused by environmental stresses, pests, and management decisions (e.g., replanting or fallowing) from

overwintering to maturity. It should also be noted that the observed difference between "planted" and "harvested" areas cannot be directly equated with precise yield losses. This difference represents an estimate influenced by uncertainties in remote sensing classification, which may include systematic errors caused by mid-season commission errors and end-season omission errors. Therefore, these results are primarily intended to reveal the spatiotemporal patterns and relative trends of potential yield reduction events rather than provide absolute production loss data. Future research could incorporate independent crop records and higher-temporal-resolution remote sensing observations to further constrain and quantify these uncertainties.

### 5.3 Limitations and future work

Despite the high spatial resolution and annual consistency achieved by the CN_Wheat10 product at the national scale, which significantly improves both the scope and accuracy of spring and winter wheat mapping, certain limitations and uncertainties remain in practical applications, particularly with regard to data completeness and regional adaptability. To enhance the stability of phenological feature extraction and the temporal continuity of the time series, this study adopted several pre-processing strategies, including cloud masking, median compositing, and linear interpolation. However, in regions frequently affected by cloud cover or with a high proportion of missing observations, the temporal continuity and availability of remote sensing imagery are still constrained. As a result, critical phenological signals during key periods may be inadequately captured, thereby affecting mapping accuracy and the spatial consistency of mapping outputs. Furthermore, in areas characterized by complex terrain and highly variable weather conditions, remote sensing observations are more prone to anomalies and noise, posing additional challenges for the accurate identification of wheat growth cycles. Although the current methodology alleviates data gaps to a certain extent, its effectiveness varies across regions, which still limits the generalizability of the product under heterogeneous environmental conditions. To enhance the applicability of CN_Wheat10 in regions with challenging topography and climatic variability, future work should focus on advancing multi-source remote sensing data fusion strategies and developing more robust temporal feature extraction and gap-filling mechanisms. Such improvements would contribute to increased stability and reliability of the dataset across diverse agroecological zones.

The feature selection in this study was based on provincial administrative units. Although it captures regional differences better than global feature selection methods, it may not completely eliminate the effects of phenological and climatic variations within provinces. Future studies can further optimize the feature selection process by using more refined agroclimatic zoning (Liu et al., 2024c) to better characterize phenological differences in wheat. In addition, while this study provides a novel multi-year, high-resolution wheat map product for China using the capabilities of Sentinel-1 and Sentinel-2, its temporal depth is inherently limited by the operational lifespan of these satellite constellations. To explore the applicability of our method over longer time series, we conducted a preliminary mapping of spring and winter wheat in Heze City, Shandong Province, for 2024 using Landsat 8 imagery, and compared the results at five representative sites with the CN_wheat10 product. The results in Fig. 20 indicate that wheat-growing areas can be identified, demonstrating cross-sensor transferability, though classification accuracy is lower than with Sentinel-2. This is primarily due to Landsat 8's coarser spatial resolution (30 m) and longer revisit

interval (16 days), which constrain cloud-free observations during key growth stages and reduce the completeness of
composites and the accuracy of time-series feature extraction. In contrast, Sentinel-2's 5-day revisit allows dense temporal
composites that capture subtle phenological dynamics. These results suggest that while Landsat 8 can support approximate
wheat mapping, achieving Sentinel-2–level precision for specific growth stages is challenging. Future integration of multi-
source satellite data could enable long-term, continuous monitoring of wheat distribution, providing insights into the dynamics
of winter and spring wheat and cropping system transitions.

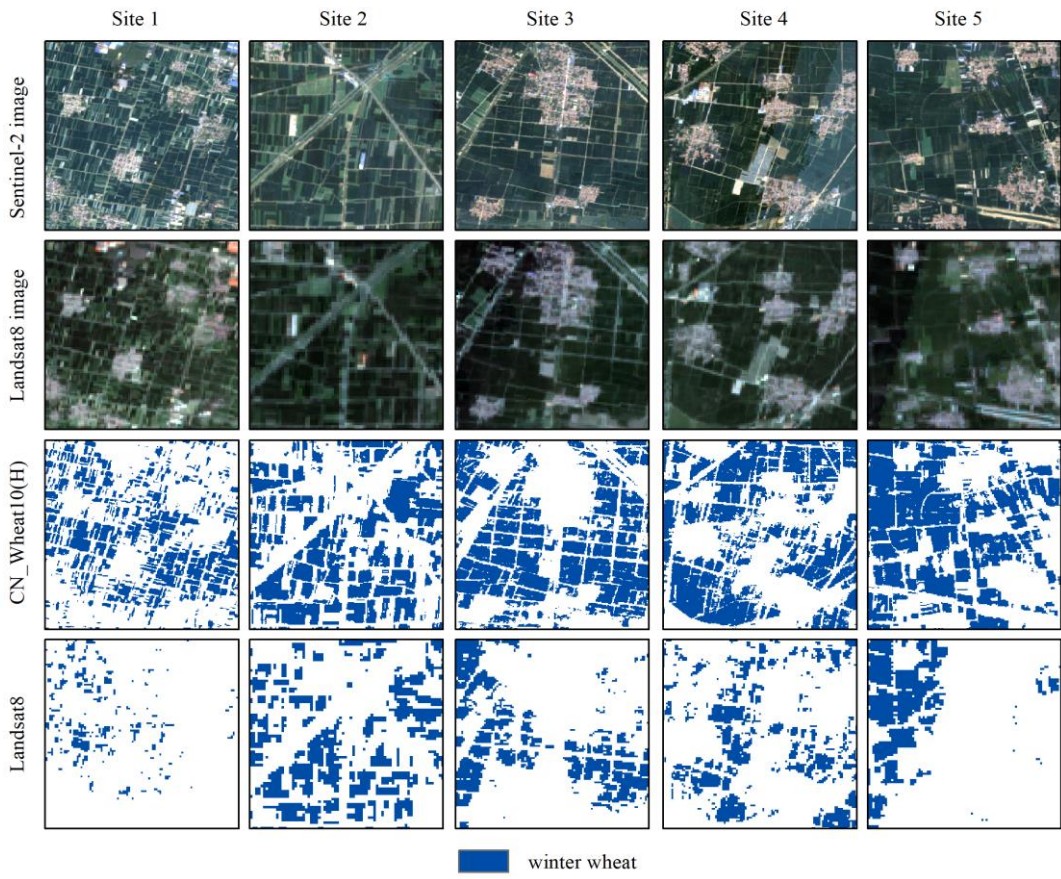


**Figure 20: Comparison of wheat remote sensing recognition regions based on sentinel-2 and Landsat8 images.**

## 6 Data availability

The CN_Wheat10 product is freely accessible at https://doi.org/10.6084/m9.figshare.28852220.v2 (Liu et al., 2025a).

## 7 Conclusions

In this study, we developed CN_Wheat10, a high-resolution (10 m) distribution product of spring-winter wheat across China for the period 2018–2024. CN_Wheat10 product includes harvested area maps for both spring and winter wheat nationwide, as well as harvested area maps for winter wheat in major producing regions, providing a comprehensive depiction of the spatiotemporal dynamics of wheat cultivation in China. Compared to existing wheat remote sensing products, CN_Wheat10 offers a key innovation by simultaneously mapping both spring and winter wheat distributions with high precision. Accuracy

assessments demonstrate that CN_Wheat10 consistently achieves high mapping performance across years and regions. For winter wheat, both planted and harvested area accuracies exceed 0.95, while spring wheat mapping during the harvested area achieves accuracies above 0.91. Additionally, comparison with official statistics (2018–2023) reveals a strong agreement, with $R^2$ values exceeding 0.94 at the provincial level and consistently above 0.88 at the municipal level. Overall, mapping performance at the planted area slightly outperforms that at the harvested area, likely due to adverse weather events such as

dry-hot wind, extreme heat, pests, and diseases, which can cause premature senescence or crop failure and reduce mapping reliability during the later growth stages. In summary, CN_Wheat10 is a high-precision, high-reliability, and high-completeness remote sensing product that integrates spatial information for both spring and winter wheat while offering detailed planted area data for core winter wheat regions. By extending the scope of wheat monitoring and enriching spatial distribution information, this product provides valuable support for agricultural monitoring, yield estimation, and disaster

response applications in China.

## Author contributions

ML designed the method, performed the analysis, wrote the manuscript and collected the validation sets. WH revised the paper. HZ revised the paper and was responsible for project management and fund acquisition.

## Competing interests

The authors declare that they have no conflict of interest.

## Financial support

This paper was supported in part by the National Key Research and Development Program of China under Grant 2022YFB3903605, and in part by the National Natural Science Foundation of China under grant numbers T2525018.

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
