# Peer review of "CN\_Wheat10: A 10 m resolution dataset of spring and winter wheat distribution in China (2018–2024) derived from time-series remote sensing"

_Earth System Science Data, 2025_

## Author Comment (AC1)

**Reviewer #1**

We thank the reviewer for a thoughtful and thorough review of our manuscript. The responses to suggestions and comments are shown in **blue** text. We have highlighted the revised sections and corresponding references in **red** text. The page (P) and line (L) numbers indicated in the response refer to the revised manuscript. The item-by-item responses to all comments are listed below.
* * *
**Suggestions and comments:**

**Point 1.** Line 174-175: "A spatially stratified sampling strategy based on quadrilateral grids was adopted to mitigate the effects of spatial autocorrelation." Could the authors specify the spatially stratified sampling strategy?

**Response:** Thank you for your valuable comment. In this study, we adopted a spatially stratified sampling strategy based on quadrilateral grids to effectively mitigate the influence of spatial autocorrelation among samples. Taking Xinyang City in Henan Province as an example (Fig. R1-1), we first divided the study area into 20km*20km quadrilateral grids on the Google Earth Engine platform and examined the monthly Sentinel-2 composite imagery from October to the following June. During the interpretation process, all land cover types were classified into six categories: winter wheat, built-up, water, trees, bare land, and other crops. Based on the proportional coverage of each land cover type within a grid, no more than 10 sample points were randomly selected per grid, thereby avoiding excessive spatial clustering of samples. Specifically, for grids containing wheat fields, 1–8 wheat samples were randomly selected in proportion to the wheat coverage within the grid, along with additional non-wheat samples to maintain category balance. For grids without wheat fields, 1–2 non-wheat samples were randomly selected for each category, including built-up land, trees, water, and bare land. This approach ensured both the spatially uniform distribution of samples and the representativeness of diverse land cover types.

**P6, L180-182 in the revised manuscript and Supplementary data:** As shown in Fig.R1-1, we take Xinyang City in Henan Province as an example to demonstrate the spatial hierarchical sampling strategy based on quadrilateral grids. We first divided the study area into a series of regular quadrilateral grids within the Google Earth Engine platform and examined the monthly Sentinel-2 composite imagery from October to the following June. During the interpretation process, all land cover types were classified into six categories: winter wheat, built-up, water, trees, bare land, and other crops. Based on the proportional coverage of each land cover type within a grid, no more than 10 sample points were randomly selected per grid, thereby avoiding excessive spatial clustering of samples. Specifically, for grids containing wheat fields, 1–8 wheat samples were randomly selected in proportion to the wheat coverage within the grid, along with additional non-wheat samples to maintain category balance. For grids without wheat fields, 1–2 non-wheat samples were randomly selected for each category, including built-up land, trees, water bodies, and bare land. This approach ensured both the spatially uniform distribution of samples and the representativeness of diverse land cover types.

[Figure]

Figure R1-1: Spatial stratified sampling strategy in Xinyang City.

**Point 2.** Line 177: "more than 50,000 valid sample points were collected annually" How much of the sample points are from the field survey? Also, how did the authors distinguish winter wheat fields from other crop types based on visual inspection? Some examples can be provided.

**Response:** We sincerely thank the reviewer for the constructive comments. Approximately 2,000–3,000 points were collected from field surveys each year during 2020–2024. These field samples were mainly concentrated in the Huang-Huai-Hai Plain and the middle and lower reaches of the Yangtze River Plain, with a smaller number collected from other provinces. The spatial distribution of these field survey samples is illustrated in Fig.R1-2.

**P6, L180-182 in the revised manuscript and Supplementary data:** Approximately 2,000–3,000 points were collected from field surveys each year during 2020–2024. These field samples were mainly concentrated in the Huang-Huai-Hai Plain and the middle and lower reaches of the Yangtze River Plain, with a smaller number collected from other provinces. The spatial distribution of these field survey samples is illustrated in Fig.R1-2.

[Figure]

Figure R1-2: Spatial distribution of sample points from field surveys in 2020-2024.

During visual interpretation of remote sensing imagery, winter wheat was distinguished from other crop types based on three main criteria: (1) Temporal dynamics (Fig.R1-3(a)): multi-temporal imagery from October to the following June reveals the complete growth cycle of winter wheat, including sowing, overwintering, regreening, jointing, heading, grain filling, and maturity, while other crops exhibit distinct temporal profiles; (2) Spectral characteristics (Fig.R1-3 (b)): winter wheat demonstrates high vegetation indices during the mid-growing season (around April), appearing as dark green in imagery, whereas rapeseed typically appears yellow and garlic is often light green; (3) Texture and spatial distribution (Fig.R1-3 (b)): winter wheat fields are generally regular in shape and spatially aggregated, in contrast to other land cover types, which often appear fragmented or irregular.

**P6, L180-182 in the revised manuscript and Supplementary data:** In the process of visual interpretation of remote sensing images, wheat is distinguished from other crop types mainly based on three criteria, taking winter wheat as an example: (1) Temporal dynamics (Fig.R1-3(a)): multi-temporal imagery from October to the following June reveals the complete growth cycle of winter wheat, including sowing, overwintering, regreening, jointing, heading, grain filling, and maturity, while other crops exhibit distinct temporal profiles; (2) Spectral characteristics (Fig.R1-3 (b)): winter wheat demonstrates high vegetation indices during the mid-growing season (around April), appearing as dark green in imagery, whereas rapeseed typically appears yellow and garlic is often light green; (3) Texture and spatial distribution (Fig.R1-3 (b)): winter wheat fields are generally regular in shape and spatially aggregated, in contrast to other land cover types, which often appear fragmented or irregular.

[Figure]

Figure R1-3: Distinguishing winter wheat from other land-cover types based on visual interpretation of remote sensing Images. (a) Phenological spectrum curve of winter wheat and remote sensing image characteristics at different growth stages. (b) Spectral texture characteristics of winter wheat and other land-cover types in April.

**Point 3**. Line 234-235: "all non-wheat pixels (Section 3.1) were classified into two types: non-wheat winter crops vs. non-winter crops and non-wheat spring crops vs. non-spring crops, according to their respective growth stages": I think there are four types?

**Response**: We sincerely thank the reviewer for this valuable comment and for pointing out the potential ambiguity in our wording. Our intention was not to classify all non-wheat pixels into four categories in a single step, but rather to conduct two separate binary classifications according to the respective growth stages. Specifically, one binary classification distinguishes between non-wheat winter crops and non-winter crops, and the other distinguishes between non-wheat spring crops and non-spring crops. These two classifications were carried out independently to match the phenological differences between winter and spring crops.

**P12, L277-280:** Specifically, for the winter growing season (October–June), the remaining non-wheat pixels were classified into winter crops (non-wheat) and non-winter crops using a binary classifier. Similarly, for the spring growing season (April–August), another binary classifier was applied to the remaining non-wheat pixels to separate spring crops (non-wheat) from non-spring crops.

**Point 4**. Line 238: What is the definition of the "spectral separability indices (SI)"?

**Response**: We thank the reviewer for this comment. The separability index (SI) is used to assess the sensitivity of the spectral separability of two classes under certain conditions, determined by the ratio of inter-class and intra-class variability. The higher the value, the better the separation between the two classes in the specified condition.

**P12, L284-286:** The SI is used to assess the sensitivity of the spectral separability of two classes under certain conditions, determined by the ratio of inter-class and intra-class variability (Somers and Asner, 2013). The higher the value, the better the separation between the two classes in the specified condition.

Somers, B. and Asner, G. P.: Multi-temporal hyperspectral mixture analysis and feature selection for invasive species mapping in rainforests, Remote Sens. Environ., 136, 14-27, https://doi.org/10.1016/j.rse.2013.04.006, 2013.

**Point 5**. Section 3.2: what would the accuracy be if you do not do the Selection of provincial feature set?

**Response**: We sincerely thank the reviewer for this constructive question. To address it, we conducted a detailed evaluation using Henan Province from 2018 to 2023 as an example. The results in Table R1-1 indicate that, without the selection of provincial feature sets, the overall accuracy ranged from 0.960 to 0.980. After applying feature selection, the accuracy improved to 0.974–0.988, with annual improvements ranging from 0.003 to 0.018. This demonstrates that, although the model already achieved relatively high accuracy without feature selection, the use of provincial feature selection further enhanced its discriminative capacity. To better illustrate this improvement, six representative locations were selected for comparison in Fig.R1-4, where it is evident that feature selection enabled more precise spatial identification of winter wheat, thereby increasing the reliability and robustness of the results.

**P26-27, L498-504:** Taking Henan Province as an example, we calculated the variation of wheat

precision before and after feature selection (Table R1-1). Without provincial feature selection, the overall precision ranged from 0.960 to 0.980; after feature selection, the overall precision increased to 0.974 to 0.988. Although the model already achieved relatively high accuracy without feature selection, the use of provincial feature selection further enhanced its discriminative capacity. To better illustrate this improvement, six representative locations were selected for comparison in Fig. R1-4, where it is evident that feature selection enabled more precise spatial identification of winter wheat, thereby increasing the reliability and robustness of the results.

Table R1-1: Comparison of wheat recognition accuracy in Henan Province before and after feature selection.

|  | 2018 | 2019 | 2020 | 2021 | 2022 | 2023 |
|---|---|---|---|---|---|---|
| No feature selection | 0.976 | 0.974 | 0.979 | 0.980 | 0.964 | 0.960 |
| Feature selection | 0.987 | 0.979 | 0.982 | 0.988 | 0.982 | 0.974 |

[Figure]

Figure R1-4: Comparison of wheat remote sensing recognition regions before and after feature selection.

**Point 6**. Line 256: It is not clear why (Yang et al. 2023) is cited here.

**Response**: We sincerely thank the reviewer for pointing out this ambiguity. Our original intention in citing Yang et al. (2023) was to reference a study that adopted a similar parameter setting for the classifier, specifically using 100 decision trees in the Random Forest algorithm while keeping other parameters at their default values. However, we acknowledge that the current wording may not clearly convey this purpose, which could lead to confusion. In order to improve the clarity, we modified the corresponding content and verified the influence of RF parameters on accuracy:

**P14, L301-304:** The classifier was implemented with 100 decision trees, there was no more significant difference in accuracy starting with 100 trees and continuing until 200 trees, as shown in Fig. R1-5. The remaining parameters were maintained at their default values, following the approach adopted in recent remote sensing studies (Yang et al., 2023).

[Figure]

Figure R1-5. Trends in accuracy of RF classifiers under different parameters at two study sites.

**Point 7**. Line 264-269: it is not clear to me why does the model can map planted winter wheat and harvested winter wheat by changing the time window of the feature set.

**Response**: We sincerely thank the reviewer for this valuable comment. The approach of using different temporal windows to map planted winter wheat and harvested winter wheat was designed with reference to previous study (Hu et al., 2024), which has demonstrated that crops at different phenological stages exhibit distinct spectral characteristics in remote sensing imagery. By selecting appropriate temporal windows, it is possible to effectively distinguish between planting and post-harvest conditions.

Specifically, winter wheat has a relatively long growth cycle of approximately 8–9 months from sowing to harvest. From early October to early April of the following year, it mainly undergoes sowing, overwintering, regreening, and jointing stages. April represents the peak growth period, during which the canopy is well developed, vegetation coverage is high, and spectral signatures are stable and distinctive, making it the optimal period for identifying the actual planting distribution. From mid- to late April, winter wheat enters the heading stage, followed by grain filling and maturity. During these stages, spectral features can be strongly influenced by adverse meteorological or biological factors, such as drought stress, high-temperature hot dry wind events, pest and disease outbreaks, and flooding. These stressors may cause premature senescence, yield reduction, or even total crop loss. Such abnormal growth events not only directly reduce the final harvested area but also lead to pronounced changes in spectral characteristics in remote sensing imagery, with a shift from the high reflectance typical of healthy vegetation to spectral patterns resembling bare soil or other non-crop surfaces, which in turn results in misclassification or exclusion during harvest-stage mapping. By late June, after harvest, most fields appear as bare soil or stubble-covered surfaces, with spectral properties markedly different from those during the planting period.

As shown in Fig.R1-6, the spectral characteristics of winter wheat at different phenological stages across three representative locations are presented, along with the corresponding differences between planted and harvested areas. This content is also described in Section 4.3 of the manuscript.

[Figure]

|  | April | May | Planted Area (CN_Wheat10(P)) | Harvested Area (CN_Wheat10(H)) | Different Area |

Figure R1-6: Spectral characteristics of winter wheat at different growth stages and differences between planted and harvested area.

Therefore, by changing the time window of the feature set, the model can extract features from different phenological phases: (1) Early October – Early April: captures the actual planted winter wheat distribution. (2) Early October – Late June: encompasses the harvest and post-harvest stages while also reflecting the impacts of adverse meteorological or biological factors on yield and final harvested area, enabling accurate mapping of harvested winter wheat.

The cited reference is as follows:

Hu, J., Zhang, B., Peng, D., Huang, J., Zhang, W., Zhao, B., Li, Y., Cheng, E., Lou, Z., and Liu, S.: Mapping 10-m harvested area in the major winter wheat-producing regions of China from 2018 to 2022, Sci. Data, 11, 1038, https://doi.org/10.1038/s41597-024-03867-z, 2024.

**Point 8**. Line 269: "The final products include harvested area maps of spring and winter wheat for 15 provinces, as well as planted area maps of winter wheat for 10 provinces." What cause the difference number of available provinces for harvested area maps and planted area maps of winter wheat?

**Response**: We sincerely thank the reviewer for this question and the opportunity to clarify. The difference in the number of provinces covered by the winter wheat planted area maps (10 provinces) and the harvested area maps (15 provinces) is mainly due to the following factors:

(1) Distribution of major winter wheat production areas: The 10 provinces for which winter wheat planted area maps were generated are located in the Huang–Huai–Hai Plain and the middle–lower reaches of the Yangtze River Plain, representing the core winter wheat-producing regions in China. These provinces have large and contiguous cultivation areas, making them suitable for accurate mapping of planted areas using remote sensing.

(2) Lower wheat cultivation intensity in other provinces: The remaining 5 provinces are mainly in northwestern China, where wheat planting is relatively limited, and fields are often small and scattered. This reduces the feasibility and reliability of extracting winter wheat planted areas at the provincial scale.

(3) Mixed cropping of spring and winter wheat: In some northwestern provinces, spring and winter wheat are cultivated in the same province. In certain areas, the harvest periods of spring and winter wheat are very close, making it difficult to reliably distinguish planted and harvested areas for wheat using remote sensing within a single growing season.

(4) Phenological differences across regions: Wheat growth cycles vary considerably across climatic zones. For example, in eastern China, winter wheat reaches its peak growth stage in April, whereas in northwestern regions, due to colder climates, winter wheat often does not reach its peak growth until May. This temporal difference, combined with overlapping growth stages and spectral characteristics of different wheat types in some regions, increases the difficulty of extracting wheat planted areas at a national scale. However, during the harvest stage, wheat exhibits distinct spectral characteristics compared to other land cover types, and the harvest period is relatively consistent across regions, allowing harvested area maps to be generated for all 15 provinces.

---

## Author Comment (AC2)

**Reviewer #2**

We thank the reviewer for a thoughtful and thorough review of our manuscript. The responses to suggestions and comments are shown in **blue** text. We have highlighted the revised sections and corresponding references in **red** text. The page (P) and line (L) numbers indicated in the response refer to the revised manuscript. The item-by-item responses to all comments are listed below.

**Suggestions and comments:**

The manuscript introduces CN\_Wheat10, a 10 m resolution dataset of spring and winter wheat distribution in China. It used a cross-regional training-sample generation method to address the lack of large-scale training data and a province-level feature-selection strategy to improve the regional adaptability of a random-forest classifier. Building on this, the authors generate separate spring- and winter-wheat maps and further derive planted- and harvested-area products, which I consider the key contribution of the work. However, it remains somewhat unclear whether the method is specifically tailored to address these mapping targets, even though the reported overall accuracies exceed 90% across years and regions. In addition, a more detailed description of the methodology and parameter settings, together with more rigorous validation, is needed to strengthen the study.

**Response**: We appreciate your considerable comments and suggestions which help to clarify the scientific significance of CN\_Wheat10 dataset and expand its applicability. We have carefully considered all of the comments and suggestions listed below and tried our best to improve the manuscript focusing on clarifying the details of the method, setting of parameters and validation.

**General comments:**

**Point 1.** One of the key contributions is mapping spring and winter wheat separately, but it is not clear how these two classes were distinguished from the outset. The authors mention using different time periods, yet it remains uncertain whether all regions were processed with distinct workflows or whether dominant spring/winter regions were predefined based on expert knowledge. The lack of overlap in the maps gives the impression that the latter approach might have been used, though this is only my inference. Since mixed-cropping areas do exist in China, clarification on this point would strengthen the manuscript.

**Response**: Thank you for this critical question, which allows us to clarify a key aspect of our methodology. Before classification, we predefined spring- and winter-wheat provinces based on agronomic expertise and long-term provincial cropping statistics, and then applied season-specific workflows and time windows accordingly. This process was not based on subjective assumptions or arbitrary decisions, but rather grounded in well-established agricultural principles derived from multiple dimensions such as long-term cropping systems, climatic conditions, accumulated temperature patterns, and sowing—harvest calendars. The goal was to ensure that the classification rules are fully consistent with actual agricultural production practices. Specifically, our approach is as follows:

- (1) Pre-definition of dominant regions: Based on extensive agronomic literature and national agricultural statistics, we first pre-defined the dominant wheat cropping systems for each province in our study area. We categorized provinces into three types: predominantly winter wheat provinces, predominantly spring wheat provinces and mixed wheat provinces.
- (2) Customize the classification process for different regions: In provinces dominated by a single crop season (e.g., Shandong—winter wheat; Qinghai—spring wheat), we used only the corresponding seasonal image time series to produce the map: winter wheat with the October to June time series; spring wheat with the April to August time series, and regard the results as the distribution of wheat in that seasonal pattern, and regard the results as the distribution of wheat in that seasonal pattern. In mixed provinces (e.g., Xinjiang—winter wheat and spring wheat), we performed two independent classification chains: one using Oct—Jun data to detect winter wheat, and one using Apr—Aug data to detect spring wheat. The two outputs were then merged using decision rules based on classifier probabilities.

The province-level predefinition based on prior agronomic knowledge was designed to make the wheat classification method more consistent with the actual agricultural planting structure. Before classification, we divided provinces into winter-wheat-dominant, spring-wheat-dominant, or mixed types based on agronomic expertise and long-term provincial cropping statistics. This process was not a subjective assumption or arbitrary decision, but was grounded in well-established agricultural principles derived from multiple dimensions, including long-term cropping systems, climatic conditions, accumulated temperature patterns, and sowing-harvest calendars. Therefore, this predefinition has a solid empirical basis and strong verifiability, effectively reflecting the objective patterns of crop growth across regions. Based on this categorization, the classification workflow can be optimized according to regional crop characteristics and seasonal signals, thereby reducing seasonal confusion and data redundancy while significantly improving classification accuracy and regional adaptability. In provinces dominated by a single wheat season, only imagery from the corresponding season was used, ensuring that the results align with local farming practices. In mixed provinces, two independent classification chains were applied and merged at the pixel level based on classification probabilities, which allows the method to accommodate the complex planting structures resulting from differences in topography or crop rotation systems. Overall, this province-level predefinition based on prior knowledge combined with season-specific classification strategy not only follows sound agricultural logic but also achieves a balance between regional consistency and classification precision, greatly enhancing the model's robustness, interpretability, and practical applicability.

To address this concern explicitly we have revised the Section 3 (Methods) to provide a much more detailed description of this multi-step workflow.

**P8, L208-217:** To distinguish spring and winter wheat, we first predefined provinces as winter-dominant, spring-dominant, or mixed based on agronomic expertise and provincial cropping statistics. Classification workflows were then tailored accordingly. In provinces dominated by a single crop season, only the corresponding seasonal time series was used: October–June of the following year for winter wheat and April–August for spring wheat. The resulting maps in these regions therefore represent only that season's wheat distribution, without overlap between spring and winter wheat. In mixed provinces, two independent classification chains were applied: one using winter-season imagery to

detect winter wheat, and the other using spring-season imagery to detect spring wheat. Pixel-level outputs were merged based on classification probabilities, when one seasonal probability was substantially higher, the pixel was assigned to that season. This "province-level predefinition plus season-specific classification" strategy ensures consistency with dominant cropping systems while adequately capturing the complexity of mixed spring—winter wheat regions.

Point 2. For the planted vs. harvested area mapping, it is unclear whether the classification was guided by specific labels or simply distinguished by using growing-season vs. full-season time series. If it is the latter, both classifiers might depend on similar mid-season features (e.g., April in Figure 4). In that case, I would consider these products to represent in-season vs. end-season maps rather than true planted vs. harvested maps. It is also unclear whether the authors considered the logical constraint that the harvested area must be a subset of the planted area. Moreover, the evaluation of wheat reduction may mask systematic errors. For example, commission errors in mid-season maps and omission errors at end-season maps could lead to large deviations in area estimates. In such cases, the difference between planted and harvested areas cannot be treated as reliable evidence of wheat reduction (Figure 15). Most importantly, area estimation should be based on rigorous statistical approaches rather than simple pixel counting.

**Response**: We sincerely thank you for these insightful and constructive comments, which have helped us identify areas where our manuscript needed greater elaboration. We have revised the manuscript accordingly and provide a point-by-point response below.

(1) With regard to the delineation of "planted area" and "harvested area," we would like to clarify that their distinction was not based on independent labels specifically tied to "planting" or "harvesting" events, but rather on time-series features derived from different phenological stages.

The key principle is that adjusting the temporal window effectively captures different agronomically meaningful phases in the wheat life cycle. After sowing, wheat must successfully overwinter and regreen to be considered as "effectively planted," while only fields that progress through heading, grain filling, and reach physiological maturity can be regarded as "harvestable." Accordingly, the "planted area" map (derived from imagery covering the overwintering to regreening period) represents wheat that has successfully established and passed through the early growth stages, whereas the "harvested area" map (extended to heading—maturity) further identifies wheat that completed critical reproductive growth and reached harvest maturity. This approach is reasonable because wheat area may shrink during the season due to management practices, environmental stress, or natural hazards, and temporal window adjustment provides a means to capture these dynamics.

We also carefully considered the terminology when naming the final products. We fully acknowledge that terms such as "mid-season map" and "end-season map" could be viewed as more neutral descriptors. Nevertheless, we opted for "planted area" and "harvested area" because these terms more directly convey the agronomic relevance of the products: the former reflects the potential cultivated extent after sowing, while the latter approximates the area that can be harvested. Compared to "mid-season/end-season," which primarily emphasize temporal positioning, "planted/harvested" explicitly relate to key crop growth stages and economic yield, thereby aligning more closely with the intended applications in agricultural monitoring and management. This naming convention is also consistent with the

terminology adopted by Hu et al. (2024). Our methodological design followed the approach proposed in Hu et al. (2024), in which different temporal windows were applied to the same training samples to construct classifiers tailored to distinct phenological stages, thereby generating both planted and harvested area maps. Using this method, Hu et al. successfully produced the ChinaWheatMap10 dataset across eight major winter wheat provinces, including ChinaWheatMap10\_P (planted area) and ChinaWheatMap10\_H (harvested area). In this study, we also compared the two types of maps to assess the consistency and differences in distribution results across phenological phases.

We recognize that the chosen terminology may invite further discussion about the nature of the approach. To ensure rigor, in the Methods and Discussion section of the revised manuscript we will clarify that "planted area" and "harvested area" can also be interpreted as phenology-based maps (i.e., in-season and end-season distributions). At the same time, we will explicitly state their agronomic meaning and associated uncertainties, so that the terminology is precise without overstating its implications.

In the current version, we have retained the original naming for the sake of continuity and consistency throughout the manuscript. However, we sincerely respect the reviewer's perspective and would be glad to consider revising the terminology in a future version if it is deemed necessary.

P14, L318-324: It is important to note that the delineation of "planted area" and "harvested area" in this study was not based on independent labels explicitly recording planting or harvesting events, but rather on adjusted temporal windows designed to capture key phenological phases of wheat growth. The maps derived from temporal window adjustment can be interpreted as phenology-based representations of winter wheat distribution. Specifically, the "planted area map" is phenologically closer to an in-season distribution, while the "harvested area map" is more comparable to an end-season distribution. Nevertheless, this correspondence should be regarded as an interpretive perspective rather than a strict equivalence to single-date mid-season or end-season classification results.

**P28, L523-533:** The "planted area" and "harvested area" maps generated in this study are essentially derived from remote sensing observations corresponding to different phenological stages. Specifically, the "planted area" represents wheat distribution identified from imagery covering the overwintering to regreening period, reflecting fields that successfully established and survived early growth, and can be interpreted as the "potential planted area." The "harvested area" is further derived by incorporating spectral information from heading to maturity on top of the planted area, aiming to capture fields that successfully completed key reproductive growth and reached a harvestable state, approximating the "actual harvested area." The choice of the terms "planted" and "harvested" is intended to more intuitively convey their agronomic relevance and maintain consistency with previous studies (Hu et al., 2024). Nevertheless, it should be emphasized that these maps can also be interpreted, in a strict sense, as "in-season" and "end-season" distributions. The differences in area between the two maps largely reflect dynamic changes in crop extent caused by environmental stresses, pests, and management decisions (e.g., replanting or fallowing) from overwintering to maturity.

**The cited reference is as follows:**

Hu, J., Zhang, B., Peng, D., Huang, J., Zhang, W., Zhao, B., Li, Y., Cheng, E., Lou, Z., and Liu, S.: Mapping 10-m harvested area in the major winter wheat-producing regions of China from 2018 to 2022, Sci. Data, 11, 1038, https://doi.org/10.1038/s41597-024-03867-z, 2024.

(2) We fully understand and agree with the logical requirement that the harvested area should be a subset of the planted area. In our methodology, the harvested area was indeed constrained within the extent of the planted area to ensure consistency with this requirement. Recognizing that our initial description may not have been sufficiently clear, we have added explicit clarification in the revised manuscript to avoid potential misunderstandings.

**P14**, **L324-325**: In addition, to satisfy the logical requirement that the harvested area should be a subset of the planted area, the harvested area in this study was masked within the extent of the planted area.

(3) We fully understand and agree with the reviewer's concerns regarding potential systematic errors. Commission errors in in-season imagery due to high vegetation coverage, as well as omission errors in end-season imagery caused by harvesting and surface changes, could indeed be amplified in difference analyses, potentially affecting interpretations of actual yield reduction. This represents a recognized limitation in our initial analysis.

In this study, the primary purpose of the "planted-harvested area difference" is to reveal the spatiotemporal patterns and relative trends of potential wheat reduction events at the national scale, rather than to provide precise estimates of absolute yield loss. In our methodological design, we incorporated certain constraints, such as masking harvested areas within the extent of planted areas and using full-season time series features to minimize cumulative errors, though we acknowledge that these measures cannot completely eliminate uncertainty.

Regarding the area estimation method, we fully recognize the importance of employing rigorous statistical approaches, as emphasized by the reviewer. In this study, pixel counting was adopted as a common and feasible approach in large-scale remote sensing mapping, particularly suited for exploratory and trend-oriented analysis. However, we agree that it has inherent limitations in terms of statistical inference.

Therefore, in the revised manuscript, we have further clarified that the planted—harvested area difference should be interpreted as an indicative and uncertain estimate, aimed at revealing the spatiotemporal distribution and relative magnitude of potential loss, rather than being directly treated as verified reduction data. Accordingly, we have adjusted all related references to "yield reduction" throughout the text to more cautious and neutral phrasing, in order to accurately reflect the nature and applicable scope of this metric.

**P21, L417-421:** It should be emphasized that these "planted–harvested differences" do not represent precise yield losses, but rather provide an indicative and uncertainty-prone measure to reveal the potential spatiotemporal patterns and relative magnitude of wheat reduction. The observed discrepancies and interannual fluctuations highlight the sensitivity of wheat cultivation to climatic variability and natural hazards, but should be interpreted primarily as qualitative or semi-quantitative signals rather than absolute production loss estimates.

**P23**, **L440-446**: It is important to emphasize that the observed "planted area > harvested area" discrepancy does not result solely from remote sensing misclassification, but reflects potential agronomic instability and environmental stress. At the same time, this difference should be interpreted as an indicative, uncertainty-prone measure, used to reveal the potential spatiotemporal patterns and

relative magnitude of wheat reduction, rather than as a direct estimate of actual yield loss. By explicitly capturing and analyzing these differences between planting and harvesting stages, the CN\_Wheat10 product provides valuable information on abnormal crop dynamics, supporting applications such as disaster impact assessment, crop insurance verification, and agricultural policy development.

**P28**, **L533-538**: It should also be noted that the observed difference between "planted" and "harvested" areas cannot be directly equated with precise yield losses. This difference represents an estimate influenced by uncertainties in remote sensing classification, which may include systematic errors caused by mid-season commission errors and end-season omission errors. Therefore, these results are primarily intended to reveal the spatiotemporal patterns and relative trends of potential yield reduction events rather than provide absolute production loss data. Future research could incorporate independent crop records and higher-temporal-resolution remote sensing observations to further constrain and quantify these uncertainties.

**Point 3.** More elaborations are needed on sample generation, feature selection, and validation. The reason for choosing the CDL of Kansas and North Dakota should be clarified (e.g., NDVI curve comparison), along with the number of CDL-derived training samples and the generated spring/winter wheat pixels. It is also unclear whether the zone strategy followed provincial boundaries or the agroecological regions in Figure 1. If it was based on the province level, this may be problematic given phenological variance (Liu et al., 2024). Please also clarify whether the VH threshold was derived using samples independent from the validation data, and consider tuning the random forest parameters (Li et al., 2023).

Liu, Yifei, Xuehong Chen, Jin Chen, Yunze Zang, Jingyi Wang, Miao Lu, Liang Sun, Qi Dong, Bingwen Qiu, and Xiufang Zhu. 2024. "Long-Term (2013–2022) Mapping of Winter Wheat in the North China Plain Using Landsat Data: Classification with Optimal Zoning Strategy." Big Earth Data 8 (3): 494–521. doi:10.1080/20964471.2024.2363552.

Li, H., Song, X.-P., Hansen, M.C., Becker-Reshef, I., Adusei, B., Pickering, J., Wang, Li, Wang, Lei, Lin, Z., Zalles, V., Potapov, P., Stehman, S.V., Justice, C., 2023. Development of a 10-m r esolution maize and soybean map over China: Matching satellite-based crop classification with s ample-based area estimation. Remote Sensing of Environment 294, 113623. https://doi.org/10.1016/j.rse.2023.113623

**Response**: We sincerely thank the reviewer for these insightful and constructive comments, which have helped us identify areas where our manuscript needed greater elaboration. We have revised the manuscript accordingly and provide a point-by-point response below.

(1) In the revised manuscript, we have added the rationale for selecting Kansas and North Dakota in the United States, analyzed their importance in U.S. wheat production, and calculated NDVI curves to examine the similarities between the wheat growth periods in the U.S. and China. The corresponding revisions are as follows:

**P9**, **L220-234**: Kansas is the leading winter wheat–producing state in the United States, characterized by vast and contiguous winter wheat fields. North Dakota, located at the heart of the United States spring wheat belt, is highly representative of spring wheat systems in terms of cropping patterns and

management practices. Both states lie in the mid-latitude region of the United States, where the photoperiod and thermal conditions are comparable to those of China's major wheat-growing zones, resulting in a strong alignment of growing seasons and phenological cycles. We randomly selected 200 spring wheat and 200 winter wheat sample points from the CDL, extracted their corresponding NDVI time series, and compared them with the NDVI profiles derived from field-collected wheat samples in China. As illustrated in Fig.R2-1, the phenological profile of spring wheat in North Dakota (sown in spring and harvested in late summer) closely matches that of the spring wheat regions in Northwest China (e.g., Xinjiang and Qinghai), while the phenological profile of winter wheat in Kansas (autumn sowing, winter dormancy, spring green-up, and early-summer harvest) closely resembles that of China's primary winter wheat regions the CDL data for both states are of high accuracy and reliability, making them ideal sources for generating high-quality and representative wheat samples.

Figure R2-1: Comparison of NDVI time series curves between spring and winter wheat in China and the United States.

(2) In the revised manuscript, we have added the number of training samples extracted from the CDL and the generated spring and winter wheat pixels. The corresponding revisions are as follows:

**P10, L241-245:** After applying confidence filtering, grid-based sampling, and temporal matching of imagery, 5,000 samples each in Kansas and North Dakota were generated, including 2,500 for wheat and 2,500 for non-wheat. The non-wheat category includes buildings, water, fallow land, tree, grassland, and other crops. These source-domain samples were then transferred to China region using Random Forest classifier in combination with Sentinel-2 time-series imagery, thereby generating wheat samples for the target region.

**P10, L258-260:** To ensure both regional representativeness and class balance, the number of samples in each province was determined based on a standardized grid approach, whereby each  $0.5^{\circ} \times 0.5^{\circ}$  grid cell was required to contain 500 sample points for wheat and 500 for non-wheat.

(3) In this study, feature selection was conducted at the provincial administrative level. The reviewer's concern that this approach may not fully capture the spatial heterogeneity of wheat phenology is highly professional and reasonable. We also carefully considered this issue during our study, but ultimately chose the province-level division for the following reasons:

We acknowledge that feature selection based on strict agro-ecological zoning is theoretically superior to using administrative provincial boundaries (e.g., Liu et al., 2024). Since provincial borders do not perfectly align with ideal agro-ecological zones, the selected features in some areas may not be locally optimal. However, considering the actual conditions of Chinese agriculture, there is currently no unified, fine-scale agro-regional classification system that can be directly applied to our classification framework. As shown in Figure R2-2, the boundaries of agricultural zones vary substantially across different sources. Under these circumstances, using provinces as the partitioning unit represents a practical and balanced compromise after weighing methodological complexity, operability, and computational efficiency. The primary goal of our study was to develop a feasible and efficient nationwide wheat mapping approach. Although simple and straightforward, the province-based strategy offers high efficiency and the advantage of leveraging relatively consistent provincial agricultural statistics and policy contexts. This allows the model to select the most relevant features under each province's dominant environmental conditions, thereby partially mitigating the effects of phenological and climatic variations. While not perfect, the province-level partition provides a feasible and efficient approximation that can reasonably represent the spatial distribution of wheat-growing regions across China, serving as a practical and effective solution for large-scale mapping tasks.

Figure R2-2. Different spatial patterns of different zoning results: (a)optimal zoning, (b)Köppen climate zoning and (c) wheat planting zoning. (The figure from Liu et al. (2024))

We agree that identifying the "optimal partition" is a valuable research direction, and the work of Liu et al. (2024) offers an excellent reference. We have added clarifications in the Discussion to explicitly acknowledge this limitation.

**P29**, **L555-558**: The feature selection in this study was based on provincial administrative units. Although it captures regional differences better than global feature selection methods, it may not completely eliminate the effects of phenological and climatic variations within provinces. Future studies

can further optimize the feature selection process by using more refined agroclimatic zoning (Liu et al., 2024) to better characterize phenological differences in wheat.

Liu, Y., Chen, X., Chen, J., Zang, Y., Wang, J., Lu, M., Sun, L., Dong, Q., Qiu, B., and Zhu, X.: Long-term (2013–2022) mapping of winter wheat in the North China Plain using Landsat data: classification with optimal zoning strategy, Big Earth Data, 8, 494-521, https://doi.org/10.1080/20964471.2024.2363552, 2024c.

(4) In calculating the VH threshold, we used some of the 2020 field survey data, which also contributed to the validation process. This was necessary because the threshold must be established based on actual crop-specific remote sensing signatures to effectively separate wheat from other crops. To address potential data dependency, we further evaluated the robustness of the threshold using different samples from different regions and years that represent distinct agro-ecological zones: (i) Hebei Province in 2021, a typical winter wheat—garlic intercropping area in northern China; (ii) Jiangsu Province in 2022, a region where winter wheat and winter rapeseed coexist in southern China; and (iii) Qinghai Province in 2019, a spring wheat—spring rapeseed coexistence area in northwestern China. As shown in Fig. R2-2, the threshold of –17.5 dB consistently distinguished wheat from other crops across years and regions, demonstrating its robustness and transferability.

**P10, L248-256 and Supplementary data:** In calculating the VH backscatter threshold, some of the 2020 field survey data were utilized for both threshold determination and validation.

To evaluate the robustness of the threshold, independent samples from different years and agroecological zones were further tested, including: (i) Hebei Province in 2021, representing a typical winter wheat-garlic intercropping area in China; (ii) Jiangsu Province in 2022, representing a region where winter wheat coexists with winter rapeseed in China; and (iii) Qinghai Province in 2019, representing a spring wheat-spring rapeseed coexistence area in northwestern China. The results demonstrate that the threshold of –17.5 dB consistently distinguished wheat from other crops across various years and regions, confirming its robustness and transferability (Fig. R2-3).

Figure R2-3. Distribution of VH backscatter values of different crops in Jiangsu Province (2021), Hebei Province (2022) and Qinghai Province (2019).

(5) Based on your suggestion, we adjusted the parameters of the random forest. We set the number of trees from 1 to 200 and tested the trend of accuracy at two different study sites. As shown in Fig. R2-4, there was no more significant difference in accuracy starting with 100 trees and continuing until 200 trees. Therefore, it is reasonable for us to set the number of trees to 100 in our study. We also added the following accuracy trends to the revised manuscript.

**P14, L301-304 and Supplementary data:** The classifier was implemented with 100 decision trees, there was no more significant difference in accuracy starting with 100 trees and continuing until 200 trees, as shown in Fig. R2-4. The remaining parameters were maintained at their default values, following the approach adopted in recent remote sensing studies.

Figure R2-4. Trends in accuracy of RF classifiers under different parameters at two study sites.

**Specific comments:**

**Point 1.** The manuscript provides harvested area maps for 15 provinces but planted area maps only for 10 provinces. What explains this discrepancy, and why were spring wheat planted area maps not produced given the similar workload and methodology?

**Response**: We sincerely thank the reviewer for this question and the opportunity to clarify. The difference in the number of provinces covered by the wheat planted area maps (10 provinces) and the harvested area maps (15 provinces) is mainly due to the following factors:

- (1) Distribution of major winter wheat production areas: The 10 provinces for which winter wheat planted area maps were generated are located in the Huang-Huai-Hai Plain and the middle-lower reaches of the Yangtze River Plain, representing the core winter wheat-producing regions in China. These provinces have large and contiguous cultivation areas, making them suitable for accurate mapping of planted areas using remote sensing.
- (2) Lower wheat cultivation intensity in other provinces: The remaining 5 provinces are mainly in northwestern China, where wheat planting is relatively limited, and fields are often small and scattered. This reduces the feasibility and reliability of extracting winter wheat planted areas at the provincial scale.
- (3) Mixed cropping of spring and winter wheat: In the northwestern regions, both winter and spring wheat are cultivated. Spring wheat has a shorter growing season, smaller sown area, and greater interannual variability compared to winter wheat. Due to these uncertainties and the less stable spatial distribution, we chose not to produce separate planted area maps for spring wheat. Instead, its distribution is reflected in the harvested area maps.

Accordingly, the planted area maps of 10 provinces emphasize the core wheat production zones, while the harvested area maps of 15 provinces provide a comprehensive overview of both winter and spring wheat distribution across China.

**Point 2.** Line 235: I found it difficult to understand the logic behind classifying non-wheat pixels into two types. This part could be explained more clearly.

**Response**: We sincerely thank the reviewer for the constructive comments. Winter crops mainly grow from October to June of the following year, while spring crops are mainly grown from April to August. Based on the clear differences in crop growth cycles, we designed two separate processes to distinguish between winter and spring crops. Specifically, for the winter growing season (October–June), winter wheat pixels were first extracted in Section 3.1. The remaining non-wheat pixels were then classified into winter crops (non-wheat) and non-winter crops using a binary classifier. Similarly, for the spring growing season (April–August), after spring wheat pixels were identified, another binary classifier was applied to the remaining non-wheat pixels to separate spring crops (non-wheat) from non-spring crops.

P12, L273-280: Winter crops mainly grow from October to June of the following year, while spring crops are mainly grown from April to August. Based on the clear differences in crop growth cycles, we designed two separate processes to distinguish between winter and spring crops. In Section 3.1, we have distinguished between spring wheat and winter wheat pixels, and the remaining non-wheat pixels are processed based on the Winter Crop Index (WCI) and automatic thresholding methods (Otsu algorithm). Specifically, for the winter growing season (October–June), the remaining non-wheat pixels were classified into winter crops (non-wheat) and non-winter crops using a binary classifier. Similarly, for the spring growing season (April–August), another binary classifier was applied to the remaining non-wheat pixels to separate spring crops (non-wheat) from non-spring crops.

**Point 3.** Line 245: Why were SAR features not included in the feature selection process but instead added only after filtering the spectral features?

**Response**: Thank you for raising this insightful question. The reason why SAR features were not included in the initial feature selection process with the optical features is based on the fundamental differences in their physical nature and data structures. We aimed to avoid potential biases that could arise from a direct comparison of these heterogeneous data sources. Our rationale is detailed below:

- (1) Data Heterogeneity and Scale Issues: Sentinel-2 spectral features (bands and indices) and Sentinel-1 SAR backscatter coefficients (VH, VV) differ significantly in their physical meaning, data distribution, and value scales. Optical features reflect the spectral reflectance properties of crops, while SAR features are sensitive to the geometric structure and dielectric properties. Performing feature selection on all 27 features together using statistical measures could introduce bias due to these vast distributional differences.
- (2) Ensuring Fairness in Feature Selection: Our primary goal was to identify the most separable optical features first. With a large number of optical features (25) that are often highly correlated, feature selection is a standard procedure to reduce redundancy. In contrast, we have only two SAR features, which provide complementary information independent of optical data. If merged into a single selection process, these two SAR features risked being "washed out" by the plethora of optical features or being misinterpreted due to their unique distribution. Our "two-stage" approach ensures that the unique information provided by SAR is fully incorporated into the classifier. This guarantees that the SAR data contributes without the risk of being prematurely eliminated in the first filtering step.

(3) Physical Interpretability: Our method follows a logic of "comparing within domain, fusing across domains." We first identify the best representatives within the optical domain and then introduce independent information from the radar domain. The resulting feature set (5 optical + 2 SAR) is more physically interpretable: it allows us to identify which spectral features are most sensitive to wheat, while also being able to ensure the incremental contribution of the SAR data on top of the optical model.

In conclusion, our intention was not to undervalue the SAR features but to prevent potential bias from a combined selection process, thereby leveraging the unique and valuable information from SAR data more fairly and effectively. Thank you again for your thorough review and valuable comments.

**Point 4.** Figure 8: The y-axis could be adjusted to a more appropriate scale to make the accuracy values easier to interpret, for example by starting at a value higher than 0.6.

**Response**: Thank you for this constructive suggestion. We agree that adjusting the y-axis scale is an effective way to enhance the clarity and interpretability of the accuracy comparison plot, as it would magnify the visual differences between the accuracy values of different classification methods.

As shown in Fig. R2-5, we have revised the figure by setting the lower limit of the y-axis to 0.8 instead of 0.6. As you anticipated, this adjustment successfully "zooms in" on the range of accuracy values, making the performance differences between the models much more apparent and easier for readers to interpret. This change undoubtedly improves the visual quality and scientific presentation of the figure.

The corresponding content has been made on Line 366 in Page 17 of the revised manuscript.

Figure R2-5: The mapping accuracy for spring and winter wheat from 2018 to 2024.

**Point 5.** Figure 9: Does the y-axis represent average overall accuracy or another metric?**

**Response**: Thank you for your comment. The y-axis in this figure represents the Average Overall Accuracy. We acknowledge that the labeling in the original manuscript was not sufficiently precise, as it omitted the crucial aspect of the metric, which could lead to ambiguity.

We have revised the figure as suggested. As shown in Fig. R2-6, the y-axis is now explicitly labeled as "Average Overall Accuracy" to ensure it is clear and unambiguous for readers.

The corresponding content has been made on Line 385 in Page 18 of the revised manuscript.

Figure R2-6: The average overall accuracy of wheat at the provincial level from 2018 to 2024.

**Point 6.** Since the temporal range of the current dataset is limited by Sentinel-2/1 data availability, is there potential to extend the mapping to longer time series using other sensors? This could be worth mentioning in the discussion. In particular, examining long-term dynamics of spring and winter wheat distribution is scientifically important for understanding cropping system shifts and their adaptation to climate change.

**Response**: We sincerely thank you for this insightful and constructive comment. We fully agree that extending the mapping time series to longer historical periods is of critical scientific importance for understanding cropping system transitions and their adaptation to climate change.

As the reviewer pointed out, the temporal range of the current dataset is indeed constrained by the availability of Sentinel-2/1 data. To explore the potential of extending the time series using other sensors, we have added an experiment in the Discussion section to preliminarily evaluate the applicability of our method using Landsat 8 imagery. Specifically, we selected Heze City in Shandong Province for 2024 as a case study and conducted a comparison with the CN\_wheat10 product at five representative sites.

The results in Fig. R2-7 indicate that wheat-growing areas can be effectively identified using Landsat 8 imagery, demonstrating the cross-sensor transferability of our approach. However, the classification accuracy is slightly lower than that achieved with Sentinel-2. This is mainly due to the coarser spatial resolution (30 m) and lower temporal frequency (16-day revisit) of Landsat 8, which limits the availability of cloud-free observations during key phenological stages, affecting the completeness of composite images and the accuracy of time-series feature extraction. This limitation is especially pronounced in regions with frequent cloud cover. In contrast, Sentinel-2's 5-day revisit cycle provides sufficient data to generate high-temporal-resolution interpolated composites (e.g., every 10 days), allowing precise capture of subtle phenological changes during each critical growth stage. Therefore, under the current Landsat data constraints, it is difficult to achieve wheat mapping at specific

phenological stages with accuracy comparable to that of Sentinel-2.

Nevertheless, this experiment demonstrates that our method is transferable across optical sensors and has the potential for historical extension. Future work will focus on developing a robust approach that integrates multi-source satellite data to overcome the limitations of single-sensor temporal continuity, thereby enabling accurate assessment of long-term spatial dynamics of winter and spring wheat distribution in China.

P29-30, L558-572: In addition, while this study provides a novel multi-year, high-resolution wheat map product for China using the capabilities of Sentinel-1 and Sentinel-2, its temporal depth is inherently limited by the operational lifespan of these satellite constellations. To explore the applicability of our method over longer time series, we conducted a preliminary mapping of spring and winter wheat in Heze City, Shandong Province, for 2024 using Landsat 8 imagery, and compared the results at five representative sites with the CN\_wheat10 product. The results in Fig. R2-7 indicate that wheat-growing areas can be identified, demonstrating cross-sensor transferability, though classification accuracy is lower than with Sentinel-2. This is primarily due to Landsat 8's coarser spatial resolution (30 m) and longer revisit interval (16 days), which constrain cloud-free observations during key growth stages and reduce the completeness of composites and the accuracy of time-series feature extraction. In contrast, Sentinel-2's 5-day revisit allows dense temporal composites that capture subtle phenological dynamics. These results suggest that while Landsat 8 can support approximate wheat mapping, achieving Sentinel-2-level precision for specific growth stages is challenging. Future integration of multi-source satellite data could enable long-term, continuous monitoring of wheat distribution, providing insights into the dynamics of winter and spring wheat and cropping system transitions.

Figure R2-7: Comparison of wheat remote sensing recognition regions based on sentinel-2 and Landsat8 images.

**Point 7.** The current validation for spring wheat is not sufficient, and the WorldCereal dataset with spring cereals map could be considered to help compensate for this gap (Van Tricht et al., 2023).

Van Tricht, K., Degerickx, J., Gilliams, S., Zanaga, D., Battude, M., Grosu, A., Brombacher, J., Lesiv, M., Bayas, J.C.L., Karanam, S., Fritz, S., Becker-Reshef, I., Franch, B., Mollà-Bononad, B., Boogaard, H., Pratihast, A.K., Koetz, B., Szantoi, Z., 2023. WorldCereal: A dynamic open-source system for global-scale, seasonal, and reproducible crop and irrigation mapping. Earth System Science Data 15, 5491–5515. https://doi.org/10.5194/essd-15-5491-2023

**Response**: Thank you for this valuable suggestion. We agree that the initial validation for spring wheat could be strengthened. In response to your comment, we have conducted an additional comparative analysis using the WorldCereal spring cereals map (Van Tricht et al., 2023) to compensate for this gap.

As shown in Fig. R2-8, we selected four representative sites and performed a visual comparison between our CN\_Wheat10 spring wheat map for 2021 and the WorldCereal spring cereals map. Our findings are as follows: There is a strong general agreement in the spatial distribution of the major spring wheat areas identified by both products. Both maps show high spatial consistency with the actual spring wheat planting patterns observable in the corresponding Sentinel-2 imagery, confirming the reliability of the identified regions. Our CN\_Wheat10 product demonstrates superior performance in capturing fine-grained details in some areas. Specifically, it delineates spring wheat field boundaries more regularly and coherently, and can even clearly distinguish subtle features like field ridges and roads, highlighting the potential advantages of our methodology in terms of mapping precision.

P15, L339-345: In Sites 1–4, we compared the CN\_Wheat10 with the WorldCereal spring cereal map in 2021. The results showed that the identified wheat areas were largely consistent between the two products and exhibited high spatial agreement with wheat-growing regions observed in Sentinel-2 imagery. In some regions, CN\_Wheat10 delineated spring wheat fields more precisely, with clearer representation of field boundaries and roads. In Sites 5–8, we compared the CN\_Wheat10 with the ChinaCP-Wheat10m spring wheat map in 2020. The ChinaCP-Wheat10m results exhibited excessive noise, blurred field boundaries, and poor spatial continuity, whereas CN\_Wheat10 demonstrated superior classification performance and spatial consistency, particularly in clearly distinguishing spring wheat from bare land and non-cropland.

Figure R2-8: Comparison of wheat details between CN\_Wheat10 products and existing published products.

---

## Author Response (AR2)

**Responses to editor and reviewers**

We sincerely appreciate the constructive comments which led us to improve the quality of this manuscript (ESSD-2025-326: CN_Wheat10: A 10 m resolution dataset of spring and winter wheat distribution in China (2018–2024) derived from time-series remote sensing). The responses to suggestions and comments are shown in **blue** text. We have highlighted the revised sections and corresponding references in **red** text. The page (P) and line (L) numbers indicated in the response refer to the revised manuscript. The item-by-item responses to all comments are listed below.
* * *
**Reviewer #2**

This paper has been improved based on the previous comments. However, several issues still need to be addressed before publication.

**Response**: We sincerely thank the reviewer for positive assessment that our manuscript has been improved and for continued engagement in providing further constructive feedback. We have carefully studied all points and have made revisions to the manuscript to address them. Our point-by-point responses to each specific comment are detailed below.

**Specific comments:**

**Point 1.** A more explicit workflow of the classification procedure is needed. The existing Figure 2 focuses primarily on feature selection and fails to convey the complete process. The overall workflow is difficult to follow, particularly given the updated classification steps involving the predefined spring- and winter-wheat-dominated provinces and the temporal windows applied to different cases.

**Response**: Thank you for this valuable comment. We agree that the previous workflow diagram did not sufficiently illustrate the complete classification process. In the revised manuscript, we have updated Fig.R1 to present a clearer and more comprehensive workflow. Specifically, we have incorporated the predefined spring- and winter-wheat-dominated provinces into the sample generation stage, ensuring that the classification logic related to different wheat-growing regions is explicitly shown. In addition, we have added schematic illustrations within the feature-selection section to clarify the distinct feature-selection strategies for spring and winter wheat. These revisions make the entire workflow easier to follow and more accurately reflect the updated classification steps described in the manuscript.

The corresponding content has been made on Line 205 in Page 8 of the revised manuscript.

[Figure]

Figure R1. Flowchart for mapping annual wheat distribution.

**Point 2.** The validation section (Figures 8, 9, and 10) lacks consistency. The comparison with WorldCereal is presented only visually, without any quantitative accuracy assessment. The same issue applies to the planted and harvested maps from ChinaWheatMap10, which were compared visually but not evaluated for accuracy in Figure 10. It should also be noted that WorldCereal includes both spring and winter cereals, whereas Table 1 and the validation focus solely on spring cereals.

**Response**: We thank the reviewer for the thoughtful and constructive comments. In the revised manuscript, we have improved the consistency of the validation section (Fig. R2-R4). Specifically, we have updated all three figures so that they now include every product used for comparison. We have also added quantitative accuracy assessments for both WorldCereal and the planted/harvested maps from ChinaWheatMap10, addressing the reviewer's concern about the previous reliance on visual comparison only. In addition, we

have incorporated winter cereals from WorldCereal into Table R1 and the validation analysis, ensuring that the products listed in Table R1 are fully aligned with those included in Fig.R2-R4. These revisions make the validation more consistent, comprehensive, and comparable across all datasets.

The corresponding content has been made in Page 16 of the revised manuscript.

[Figure]

Figure R2: Comparison of wheat details between CN_Wheat10 products and existing published products.

The corresponding content has been made in Page 17 of the revised manuscript.

[Figure]

Figure R3: The mapping accuracy for spring and winter wheat from 2018 to 2024.

The corresponding content has been made in Page 18-19 of the revised manuscript.

The Fig.R4 shows that ChinaCP-Wheat10m and WorldCereal achieve relatively high accuracies in certain provinces. This is largely because both products provide wheat distribution maps for a single year, reflecting the accuracy for that specific year only, whereas the other products report multi-year average accuracies.

[Figure]

Figure R4: The average overall accuracy of wheat at the provincial level from 2018 to 2024.

The corresponding content has been made in Page 7 of the revised manuscript.

Table R1: Information on the reference wheat mapping products used for comparison.

| Wheat maps | Wheat types | Study area | Resolution | Time range | Reference |
|---|---|---|---|---|---|
| ChinaWheat10 | winter wheat | 11 provinces | 10 m | 2018-2024 | (Yang et al., 2023) |
| ChinaWheatMap10 | winter wheat | 8 provinces | 10 m | 2018-2022 | (Hu et al., 2024) |
| ChinaCP-Wheat10m | spring and winter wheat | China | 10 m | 2020 | (Qiu et al., 2025) |
| WorldCereal | spring and winter cereals | Global | 10 m | 2021 | (Van Tricht et al., 2023) |
| TWDTW_Map | winter wheat | 11 provinces | 30 m | 2001-2023 | (Dong et al., 2020) |

Note: ChinaWheatMap10 includes planted area maps (ChinaWheatMap10_P) and harvested area maps (Chinawheatmap10_H). The last product was generated by TWDTW algorithm, we call this product TWDTW_Map for short.

**Point 3.** Area estimation should be based on statistical approaches rather than simple pixel counting, and the associated uncertainty needs to be discussed in Section 4.2 & 4.3.

(See CEOS WGCV LPV Land Cover Validation Protocol, 2025. Available at: https://lpvs.gsfc.nasa.gov/PDF/CEOS_WGCV_LPV_Land_Cover_protocol_Sept2025_V1.pdf)

**Response**: We thank the reviewer for this constructive comment. In the revised manuscript, we performed calculations using a statistically based area estimation method following good practice recommended by Olofsson et al.(2014). Specifically, we now estimate class areas using sample-based estimators derived from the confusion matrix, and report the associated standard errors and 95 percent confidence intervals. We have also added a detailed discussion of area estimation uncertainty in Sections 4.2 and 4.3. These revisions strengthen the rigor of the area estimation procedure and address the reviewer's concern.

[revised manuscript text omitted]

**Point 4.** The authors explained in the response the differences between the number of available provinces for the harvested area maps and the planted maps. This point is worth mentioning in the manuscript as well.

**Response**: Thank you for your helpful suggestion. We agree that the difference in the number of provinces available for the harvested area maps and the planted area maps is an important point that should be clarified in the manuscript. In the revised version, we have added an explicit explanation to the study area description section. The new text states that the ten provinces in eastern and southern China have large and contiguous winter wheat cultivation areas, which allows both planted and harvested area maps to be generated, whereas the five provinces in northern and northwestern China have limited and fragmented wheat cultivation, with mixed spring and winter wheat planting and similar harvest periods, so only harvested area maps were produced for these regions. This addition ensures that the rationale is clearly communicated in the main text.

**P5, L135-141:** Among the 15 provinces included in this study, the ten provinces located in eastern and southern China, which encompass the Huang–Huai–Hai Plain and the middle and lower reaches of the Yangtze River Plain, constitute the country's core winter wheat production regions. These areas are characterized by large cultivation scales and highly contiguous fields, which enables the extraction of both planted area maps and harvested area maps. In contrast, in the five provinces located in northern and northwestern China, wheat cultivation is relatively limited and fragmented, and some regions involve mixed planting of spring wheat and winter wheat with harvest periods that occur close to each other. As a result, only harvested area maps were generated for these provinces in this study.

**Point 5.** Figure 1(a) and Figure 6 display agro-agricultural zoning, this may lead to confusion to the fact that the paper actually follow province-level zoning

**Response**: Thank you for pointing this out. To avoid any potential confusion, we have removed the agro-agricultural zoning from the figures and replaced it with the corresponding province-level divisions. The revised content is now presented in the updated Fig. R6- R7.

The corresponding content has been made on Line 145 in Page 5 of the revised manuscript.

[Figure]

Figure R6: Location of study area in China. (a) Location of 15 provinces and 3 municipalities included in the study area. (b) Proportion of wheat production in 2022.

The corresponding content has been made in Page 13 of the revised manuscript.

[Figure]

Figure R7: NDVI curves for different land cover types in four provinces.

**Point 6.** Please clarify how Figure 4 was generated. Confirm whether the samples used for VH analysis are independent from those used for validation.

**Response**: We thank the reviewer for raising these points, which we agree are crucial for clarity. Please find our clarifications below:

**Generation of Figure 4:** This figure was created to analyze the distribution of different land cover types within the derived wheat probability map. For each class, we randomly selected 500 sample points from the field survey data and plotted their frequency distribution against the wheat probability values.

**P10, L241-243:** To analyze the distribution of different land-cover types within the derived wheat probability map, we randomly selected 500 sample points for each land-cover category from the field survey data and plotted their frequency distribution against the corresponding wheat probability values.

**Independence of Samples:** We confirm that the samples for the VH analysis were indeed a randomly selected subset of the broader validation samples. This approach was taken to ensure the representativeness of the analysis while using our existing high-quality ground truth data. We acknowledge that they are not an independent set, and this point is now explicitly stated in the revised text to prevent any potential misunderstanding.

**P10, L246-248:** It should be noted that the sample set used for this VH analysis constitutes a randomly selected subset of the overall validation samples. This approach ensures the representativeness of the analysis while using the existing high-quality ground-truth data.

**Point 7.** Figure 19 requires more explanation in the caption (e.g., acquisition time, types of misclassified land cover).

**Response**: Thank you for the constructive suggestion. We have expanded the caption of Fig.R8 to include additional details such as the acquisition time of the imagery and the non-wheat area that were misclassified.

The corresponding content has been made in Page 30 of the revised manuscript.

[Figure]

Figure R8: Comparison of wheat remote sensing recognition regions before and after feature selection.

**Point 8.** Line 225: The detailed explanation on how Figure 3 was generated could be moved to its caption to make the text more concise.

**Response**: We thank the reviewer for this valuable suggestion. We have moved the detailed explanation regarding the generation of the Figure from the manuscript to the figure caption to enhance the conciseness of the text. The revised caption now fully describes the data sources and methodology, as shown below.

**P9, L219-228:** Kansas and North Dakota are representative of winter and spring wheat systems, respectively, and their mid-latitude conditions result in strong phenological alignment with China's major wheat zones. As illustrated in Fig. R9, the phenological profiles from the United States closely match those of the corresponding wheat types in China, confirming the representativeness of the constructed samples.

[Figure]

Figure R9: Comparison of NDVI time series curves between spring and winter wheat in China and the United States. (a) Spring wheat NDVI profile from field survey data in Northwest China. (b) Spring wheat NDVI profile from CDL data in North Dakota, USA. (c) Winter wheat NDVI profile from field survey data in the eastern plains of China. (d) Winter wheat NDVI profile from CDL data in Kansas, USA.

**Point 9.** Please explain how the number of samples for training and validation was determined, whether it is supported by any reference, and whether the sampling design is reasonable.

Tyukavina, A., Stehman, S. V., Pickens, A. H., Potapov, P., & Hansen, M. C. (2025). Practical global sampling methods for estimating area and map accuracy of land cover and change. Remote Sensing of Environment, 324, 114714. https://doi.org/10.1016/j.rse.2025.114714.

**Response**: Thank you for your valuable comment. Your question regarding the basis for determining the numbers of training and validation samples, as well as the overall reasonableness of the sampling design, is essential for ensuring the rigor of our study. Below we provide a detailed explanation of the procedures used to determine the training and validation sample sizes

**Determination of the training sample size:** The number of training samples was established using a data-driven and empirically based accuracy convergence analysis. Specifically, within each 0.5° grid cell, we progressively increased the number of randomly generated sample points from 20 to 1000 and systematically evaluated how classification accuracy changed with increasing sample size. The results indicated that when the numbers of wheat and non-wheat samples within each grid reached approximately 500 each, the model accuracy stabilized and no longer showed substantial improvement with additional samples. Therefore, we selected 500 wheat samples and 500 non-wheat samples as the most appropriate training sample size.

This choice ensures strong model performance while avoiding unnecessary computational costs. The relevant description is provided in the original manuscript on page 10, lines 257 to 261, as well as in the supplementary materials.

**P10, L257-261:** To ensure both regional representativeness and class balance, the number of samples in each province was determined based on a standardized grid approach, whereby each 0.5° × 0.5° grid cell was required to contain 500 sample points for wheat and 500 for non-wheat. This design effectively supports the robustness and generalizability of the classification model across heterogeneous agro-ecological zones. The sample size selection process is shown in Fig. R10.

[Figure]

Figure R10. Changes in accuracy of winter wheat mapping under different sample sizes. (a)Variation trend of wheat accuracy in Shandong (SD) province. (b)Variation trend of wheat accuracy in Hubei (HuB) province.

**Sampling design for validation samples:** The sampling design for validation samples was developed with reference to methods used in previous studies (Liu et al., 2024; Liu and Zhang, 2023) and was adjusted to suit the specific characteristics of our research area. This approach ensures both scientific rigor and methodological appropriateness. We first divided the study area into regular quadrilateral grids, which served as the basic spatial units for sampling. All sample points were then manually interpreted using monthly Sentinel-2 composite images from the growing season, and were assigned to one of six land-cover categories: wheat, built-up land, water, trees, bare land, and other crops. For sample allocation, we applied a combined strategy that incorporates proportional allocation and category balancing. In grids containing wheat, we randomly selected between one and eight wheat samples according to the proportional wheat coverage within each grid, and supplemented these with non-wheat samples to maintain category balance. In grids without wheat, we randomly selected one to two non-wheat samples for each of the four categories, namely built-up land, trees, water, and bare land. The total number of samples per grid was limited to no more than ten, which effectively reduces excessive spatial clustering and minimizes the influence of spatial autocorrelation on the validation results.

This sampling strategy ensures a spatially uniform distribution of samples and provides representative coverage of diverse land-cover types, while also adhering to the fundamental principles of stratified sampling in statistics. A detailed description of this method is provided in the manuscript on page 6, lines 175 to 179, as well as in the supplementary materials. To further enhance the rigor of the study, the corresponding literature references have been added in the revised version.

**P6, L175-181:** The sampling design of validation samples refers to the methods in previous studies (Liu et al., 2024b; Liu and Zhang, 2023), and has been adjusted according to the specific conditions of this study to

ensure its scientific and rationality. Multi-temporal Sentinel-2 imagery from 2017 to 2024 was dynamically explored through the Google Earth Engine (GEE) visualization platform. Manual interpretation was conducted by combining spectral, textural, and temporal variation characteristics. A spatially stratified sampling strategy based on quadrilateral grids was adopted to mitigate the effects of spatial autocorrelation. To further improve interpretation accuracy and boundary delineation, historical very high-resolution imagery (GE-VHR) from Google Earth was used for auxiliary verification.

Liu, W., Li, S., Tao, J. et al. CARM30: China annual rapeseed maps at 30 m spatial resolution from 2000 to 2022 using multi-source data. Scientific Data 11.1 (2024): 356. https://doi.org/10.1038/s41597-024-03188-1.

Liu, W. and Zhang, H. Mapping annual 10 m rapeseed extent using multisource data in the Yangtze River Economic Belt of China (2017–2021) on Google Earth Engine. International Journal of Applied Earth Observation and Geoinformation 117 (2023): 103198. https://doi.org/10.1016/j.jag.2023.103198.

**P3-4 in supplementary data:**As shown in Fig.R11, we take Xinyang City in Henan Province as an example to demonstrate the spatial hierarchical sampling strategy based on quadrilateral grids. We first divided the study area into a series of regular quadrilateral grids within the Google Earth Engine platform and examined the monthly Sentinel-2 composite imagery from October to the following June. During the interpretation process, all land cover types were classified into six categories: winter wheat, built-up, water, trees, bare land, and other crops. Based on the proportional coverage of each land cover type within a grid, no more than 10 sample points were randomly selected per grid, thereby avoiding excessive spatial clustering of samples. Specifically, for grids containing wheat fields, 1–8 wheat samples were randomly selected in proportion to the wheat coverage within the grid, along with additional non-wheat samples to maintain category balance. For grids without wheat fields, 1–2 non-wheat samples were randomly selected for each category, including built-up land, trees, water bodies, and bare land. This approach ensured both the spatially uniform distribution of samples and the representativeness of diverse land cover types.

[Figure]

Figure R11: Spatial stratified sampling strategy in Xinyang City.